# mTORC1 accelerates retinal development via the immunoproteasome

Ji-Heon Choi[1], Hong Seok Jo[1,2], Soyeon Lim[1], Hyoung-Tai Kim[1], Kang Woo Lee[1], Kyeong Hwan Moon[1], Taejeong Ha[1], Sang Soo Kwak[1], Yeha Kim[1], Eun Jung Lee[1], Cheol O. Joe[1] & Jin Woo Kim [ID] [1]

The numbers and types of cells constituting vertebrate neural tissues are determined by cellular mechanisms that couple neurogenesis to the proliferation of neural progenitor cells. Here we identified a role of mammalian target of rapamycin complex 1 (mTORC1) in the development of neural tissue, showing that it accelerates progenitor cell cycle progression and neurogenesis in mTORC1-hyperactive tuberous sclerosis complex 1 (*Tsc1*)-deficient mouse retina. We also show that concomitant loss of immunoproteasome subunit Psmb9, which is induced by Stat1 (signal transducer and activator of transcription factor 1), decelerates cell cycle progression of *Tsc1*-deficient mouse retinal progenitor cells and normalizes retinal developmental schedule. Collectively, our results establish a developmental role for mTORC1, showing that it promotes neural development through activation of protein turnover via a mechanism involving the immunoproteasome.

[1] Department of Biological Sciences, Korea Advanced Institute of Science and Technology (KAIST), Daejeon 34141, South Korea. [2] Present address: MEDIMABBIO INC., Seoul 06306, South Korea. These authors contributed equally: Ji-Heon Choi, Hong Seok Jo. Correspondence and requests for materials should be addressed to J.W.K. (email: jinwookim@kaist.ac.kr)

Cells constituting the vertebrate nervous system are derived from neural progenitor cells (NPCs), which repeatedly divide to self-renew and produce various types of neurons and glia in particular developmental schedules[1, 2]. The proliferating NPC population decreases gradually during development, while post-mitotic neurons (PMNs) derived from the NPCs accumulate in neural tissues[3]. These changes not only result from a decrease in the proliferating NPC population within the tissues but also reflect slow progression of the progenitor cell cycle[3]. Therefore, alterations of cell cycle regulator expression often result in the changes in size and composition of neural tissues. For instance, the mice lacking cyclin D1 (*CcnD1*), which is

necessary for NPC to stay in cell cycle, show reduced body and organ sizes[4]. The numbers of cells comprising of *CcnD1*-deficient mouse retina were also remarkably reduced, more significantly in early-born cell types, and this was accounted by slow cell cycle progression and fast cell cycle exit of *CcnD1*-deficient NPCs[5, 6]. Concomitant loss of p27[Kip1] rescues histogenic defects in the *CcnD1*-deficient mouse retina by preventing *CcnD1*-deficient NPCs from exiting the cell cycle precociously[7]. Therefore, not only keeping NPC population in proper numbers but also maintaining NPC cell cycle at proper speed is necessary to construct nervous tissues with constant size and composition.

The mouse retina has been employed as a model for studying developmental mechanisms shared among mammalian nervous tissues[8]. Previous studies have found that retinal progenitor cells (RPCs) undergo a serial transition of cell fate during development, producing six major neuronal subtypes and Müller glia (MG) in a stage-specific manner[1, 8]. The transition is not only related to serial changes in fate determinants in RPCs; it is also linked to the changes of environment established by newly produced retinal neurons[1, 9]. The contribution of internal determinants gradually decreases, whereas external factors produced by neighboring neurons come to play greater roles over time. For instance, growth differentiation factor 11 (Gdf11) is produced by retinal ganglion cells (RGCs) and instructs neighboring RPCs to stop producing additional RGCs[10]. Sonic hedgehog is also produced by RGCs, but it maintains RPC fate[11]. Therefore, decisions of RPCs to maintain their fate or exit cell cycle to differentiate to retinal neurons are combined outputs of internal determinants and external signals.

The external factors trigger their unique intracellular signaling cascades, which control the fates of RPCs in an evolutionarily conserved manner[8, 12]. At the same time, these signals are also integrated into intracellular signaling pathways that are ubiquitously distributed and regulate cell proliferation, differentiation, and survival. The phosphoinositide 3-kinase (PI3K)-Akt signaling cascade is one of the ubiquitous intracellular signaling pathways that couple retinal neurogenesis and RPC proliferation in diverse organisms[13, 14]. *Drosophila* eye imaginal disc cells that either overexpress the PI3K catalytic subunit, p110, or lack the phosphoinositide 3-phosphatase phosphatase tensin homolog (PTEN), hyperproliferate to produce more retinal neurons than neighboring wild-type (WT) eye disc cells[13]. Similar autonomous neurogenic acceleration has also been observed in *Pten*-deficient mouse retinas, in which hyperproliferating RPCs produce all types of retinal neurons in advance of their regular schedule[14]. In both cases, Akt-hyperactive RPCs do not simply expand themselves, but they also produce retinal neurons more rapidly than neighboring WT RPCs.

However, the Akt downstream targets responsible for the diverse changes in the retinas, including hyperproliferation, accelerated neuronal differentiation, enhanced survival, and synaptic dynamics of retinal neurons, remain unclear[13–16]. Among downstream components of PI3K-Akt pathway, the target of rapamycin (TOR) is an evolutionarily conserved serine/threonine protein kinase that interacts with two distinct sets of adaptor proteins to form the TOR complex 1 (TORC1) and complex 2 (TORC2). The mammalian TORC1 (mTORC1) regulates cell proliferation and growth, while mTORC2 mainly involve in cell shape change and motility[17]. Akt phosphorylates and inactivates tuberous sclerosis complex 2 (TSC2), the GTPase-activating protein that inhibits Ras homolog enriched in brain 1 (RHEB1) GTPase and thereby interferes with RHEB1-dependent activation of TORC1[18, 19]. The mTORC1 not only supports cell proliferation at downstream of PI3K-Akt pathway but it also senses cellular energy levels to facilitate anabolic processes, such as protein and lipid synthesis, but limits catabolic processes, such as autophagy and lysosome biogenesis[17, 20]. Hyperactivation of mTORC1 either by activating mutations of mTORC1 components or inactivating mutations of *TSC* closely related with various neurological diseases, such as brain tumors, epilepsy, and autism[21]. The evidences also demonstrate that TORC1 supports neurogenesis in the retina of *Drosophila* and zebrafish[13, 22].

In this study, we investigate the roles of mTORC1 as a downstream mediator of Akt-induced developmental changes in mouse retina. In tuberous sclerosis complex 1 (*Tsc1*)-deficient mouse retina where mTORC1 is hyperactive, all retinal cells were born ahead of their regular developmental schedules. This was related with the mTORC1-dependent acceleration of progenitor cell cycle. To expedite cell cycle progression, mTORC1 not only facilitates synthesis but also promote degradation of the proteins, including cyclins. For the latter event, mTORC1 induces the expression of immunoproteasome component Psmb9 via signal transducer and activator of transcription 1 (Stat1) in the RPC. Collectively, our results suggest a role of the PI3K-Akt-mTORC1 pathway, which promotes the development of neural tissues by coupling synthesis and degradation of cell cycle regulator proteins in the progenitor cells.

## Results

**Developmental acceleration of the *Tsc1-cko* mouse retina.** Given the hyperactivation of mTOR in the Akt-hyperactive *Pten-cko* mouse retina (Supplementary Fig. 1), we hypothesized that mTOR pathway might play a role in the PI3K-Akt-induced developmental acceleration of the mouse retina as it regulates *Drosophila* retinal neurogenesis[13]. To test this hypothesis, we

**Fig. 1** Normal cell composition but neuronal enlargement of *Tsc1-cko* mouse retina. **a** Distribution of cells underwent Cre-mediated deletion of *Tsc1* gene in E14.5 *Tsc1-cko;R26*[+/lacZ] mouse retina was visualized indirectly by immunodetection of ß-galactosidase (ß-gal), which is expressed from a *lacZ* gene at Cre-recombined *R26R* locus. Activities of mTORC1 and mTORC2 in the retinas were also measured by immunodetection of pS6 and pAkt(S473), respectively. Scale bars, 100 μm. **b** Relative levels of mTOR pathway components in the mouse retinas were examined by western blotting (WB) with antibodies against corresponding proteins. SM size marker. **c** Hematoxylin and eosin (H&E) staining images of P14 *Tsc1-het* and *Tsc1-cko* littermate mouse retinal sections. Sizes of blue and green bars in two bottom images are same. Scale bars, 100 μm. **d** P14 littermate mouse eye sections were stained with antibodies that recognize Brn3b (RGC), Pax6 (AC), Calbindin (AC subset and HZ [arrowheads]), Chx10 (BP), Rhodopsin (Rhod; rPR), green/red-opsin (G/R-opsin; cPR), and Sox9 (MG). Scale bars, 200 μm. **e** Relative numbers of cells expressing the markers in the *Tsc1-cko* retinas were obtained by comparing with those in the *Tsc1-het* retinas. Numbers of retina analyzed are 4 (from 3 independent litters). **f** HZ, rod BP, and AC cells in P14 *Tsc1-het* and *Tsc1-cko* littermate mouse retinas are visualized by immunostainings with antibodies detecting respective markers Calbindin, protein kinase C-α (PKCα), and Syntaxin. Arrowheads indicate cell bodies of those retinal neurons. **g** Average area of the neuronal cell body in P14 *Tsc1-cko* mouse retinas was compared with that of littermate *Tsc1-het* mouse retinas. Values are averages of 200 cells in 4 different mouse retinas collected from 3 independent litters. **h** (Left) P14 *Tsc1-het* and *Tsc1-cko* mouse retinal cells were analyzed by FACS to compare their relative cell sizes by measuring forward scatter (FSC) values. (Right) Relative sizes of *Chx10::GFP*-positive bipolar cells in those littermate mouse retinas were also compared by measuring FSC values. **i** Mean sizes of the cells in the *Tsc1-cko* mouse retinas were obtained and shown in a graph as relative values to *Tsc1-het* samples (*n* = 3). Error bars in the graphs denote standard deviations (SD). *P*-values are obtained by Student's *t*-test and shown in the graphs (*<0.05; **<0.01; ***<0.001)

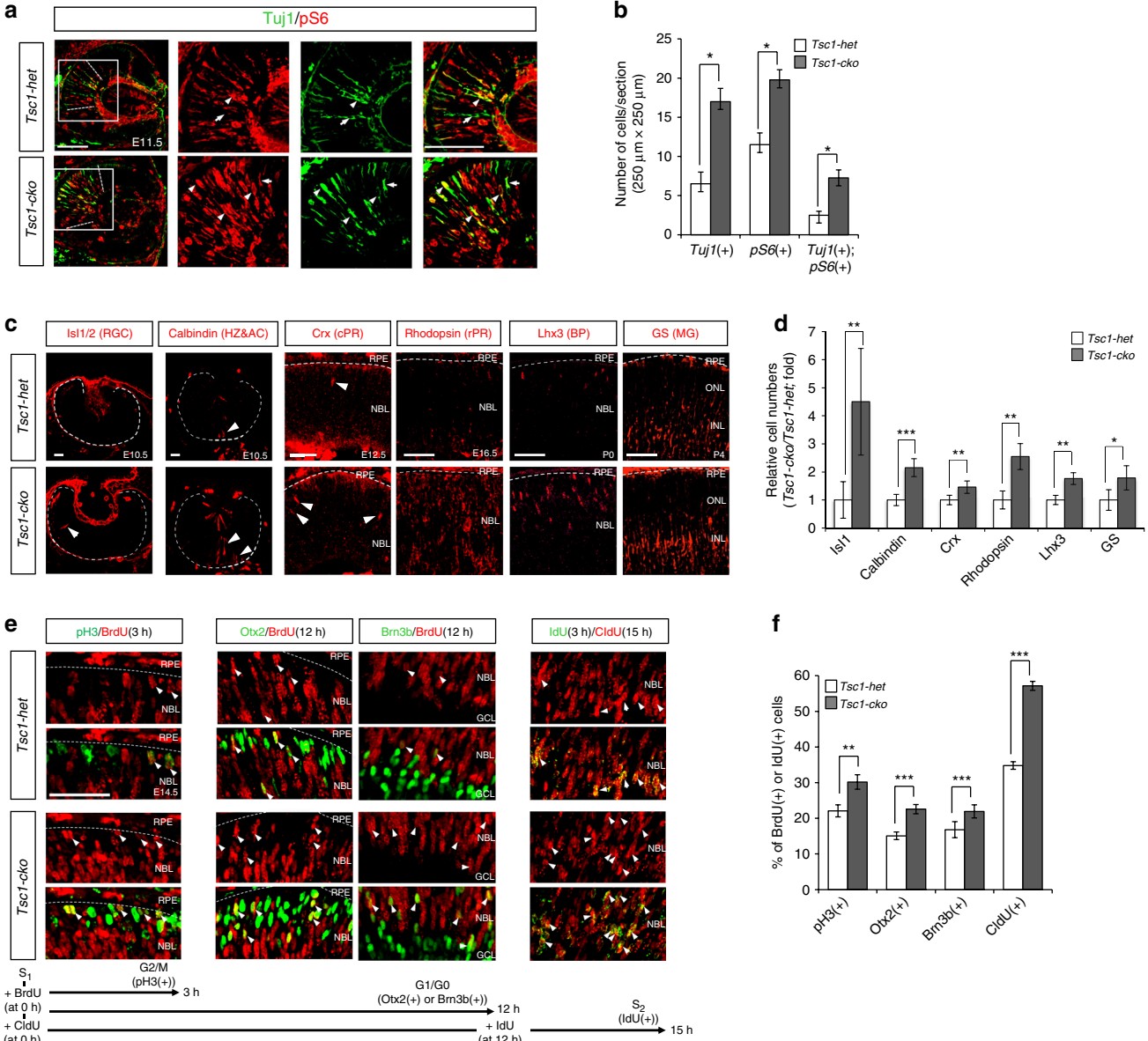

**Fig. 2** mTORC1 activation accelerates RPC cell cycle and retinal development. **a** E11.5 *Tsc1-het* and *Tsc1-cko* littermate mouse retinal sections (coronal, 12 μm) were co-stained with mouse anti-tubulin-βIII antibody (Tuj1; green) and rabbit anti-pS6 antibody (red). Images in three right columns are corresponding to the lined boxes in the left. Dotted lines indicate dorsal (top) and ventral (bottom) margins of Tuj1-positive cell population. Scale bar, 100 μm. **b** Numbers of stained retinal cells in an area (250 μm × 250 μm) were shown in a graph. Numbers (*n*) of retinas analyzed are 5 obtained from 3 independent litters. **c** Distribution of retinal neurons in littermate *Tsc1-het* and *Tsc1-cko* mouse retinas at corresponding time points, when each retinal subtype is produced prominently, was analyzed by immunostainings of various cell type-specific markers. Dotted lines mark retina-RPE borders. Arrowhead indicates marker-positive post-mitotic neurons. Scale bar, 100 μm. **d** Relative numbers of retinal cells in the *Tsc1-cko* mice were compared with those in *Tsc1-het* mice and shown in a graph. Numbers of samples analyzed are 6 (E10.5, E12.5 and E16.5) and 4 (P0 and P4) prepared from 4 and 3 independent litters, respectively. **e** Pregnant female mice were injected with BrdU (5 mg/kg) at 3 h (for pH3 co-staining) or 12 h (for Otx2 or Brn3b co-staining) prior to the collection of embryos at 14.5 dpc. Alternatively, mice were injected with CldU and IdU at 15 and 3 h prior to the collection of the embryos, respectively. Retinal sections were then co-stained with rat anti-BrdU (red) and corresponding rabbit antibodies (green) for the former analyses, and mouse anti-CldU (green) and rabbit anti-IdU (red) antibodies for the latter analysis. Arrowheads indicate cells co-expressing BrdU and corresponding markers. Scale bar, 50 μm. **f** Ratio of BrdU-co-expressing cells to total marker-expressing cells was obtained from 5 different samples from 3 independent litters and shown in a graph. Values in the graph are average and error bars are SD. *P*-values are obtained by Student's *t*-test and shown in the graphs (*<0.05; **<0.01; ***<0.001)

generated *Tsc1^flox/flox^*;*Chx10-Cre* (*Tsc1-cko*) mice lacking Tsc1/Hamartin, which cooperates with Tsc2/Tuberin to inhibit Rheb1-dependent activation of mTORC1 specifically in the retina[18, 19, 23, 24]. Consequently, mTORC1 activity, measured indirectly by detecting the phosphorylation of ribosomal S6 protein at S235/S236 (pS6) by an mTORC1 target S6 kinase, was

greatly elevated in the *Tsc1-cko* mouse retina in comparison to *Tsc1^flox/+^*;*Chx10-Cre* (*Tsc1-het*) littermate mouse retina (Fig. 1a, b; there was no phenotypic difference among *Chx10-Cre*, *Tsc1^flox/flox^* (*Tsc1-flox*), and *Tsc1-het* mice [data not shown]). Overall size of the eye of *Tsc1-cko* mice was not different significantly from *Tsc1-het* littermates, although the retinas of *Tsc1-cko* mice were

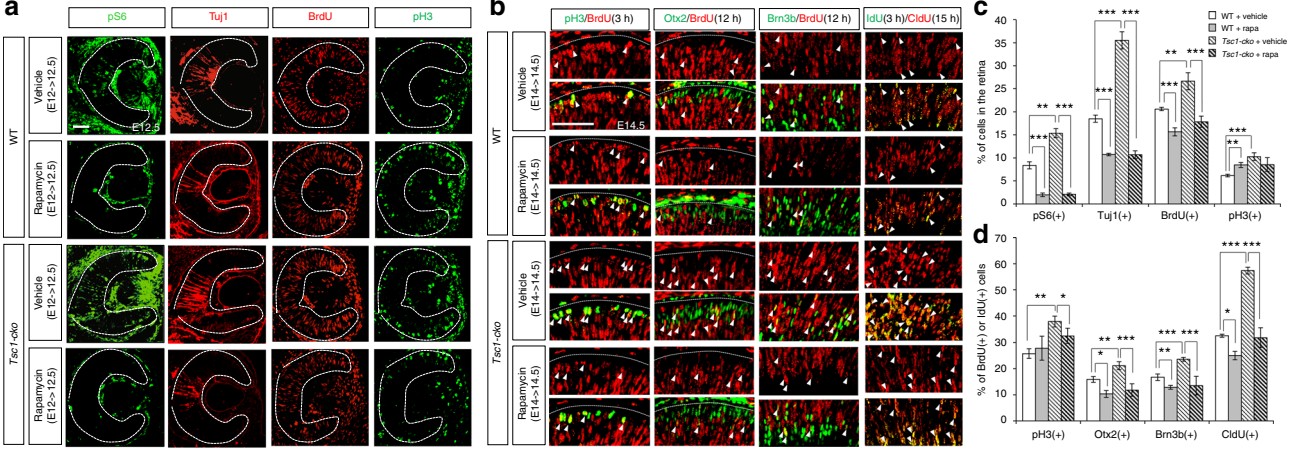

**Fig. 3** mTORC1 inhibition by rapamycin inhibits RPC cell cycle and retinal development. **a** Pregnant mice were injected with vehicle (see Methods) or rapamycin (5 mg/kg) into the peritoneum at 12 dpc, and the embryos were collected at 12.5 dpc (E12.5). BrdU were injected at 3 h prior to the embryo collection. Distribution of each marker in the embryonic retinas was investigated by immunostaining with corresponding antibodies. Dotted lines mark retinal margins. Scale bar, 100 μm. **b** Cell cycle progression of embryonic RPCs in the rapamycin-injected mice was examined as described in Fig. 2f. Scale bar, 100 μm. **c** Percentages of cells expressing the markers in **a** are shown in a graph. Values are averages (*n* = 5 from 3 independent litters) and error bars denote SD. **d** Ratios of BrdU-co-expressing cells to total marker-expressing cells are shown in a graph (*n* = 5 from 3 independent litters). Error bars denote SD. *P*-values are obtained by one-way ANOVA test and shown in the graphs (*<0.05; **<0.01; ***<0.001)

thicker than *Tsc1-het* littermate mouse retinas about 1.3-fold (Fig. 1c). Cell composition of post-natal day 14 (P14) *Tsc1-cko* mature mouse retina was not significantly different from that of *Tsc1-het* littermate retina, except for RGCs that are less in *Tsc1-cko* (Fig. 1d, e). However, mean size of cells in P14 *Tsc1-cko* mouse retina are over 1.2-fold larger than that in *Tsc1-het* littermate retina (Fig. 1f–i), suggesting that Tsc1 is important for regulating the size and morphology of retinal neurons but not their cell fates.

Using *Tsc1-cko* mice, we examined whether the loss of *Tsc1* recapitulates developmental changes, including hyperproliferation, accelerated neurogenesis, and enhanced cell survival, observed in the *Pten-cko* mouse retina[14]. First, we investigated neurogenesis in the *Tsc1-cko* mouse retina by immunostaining for neuron-specific tubulin-βIII using the Tuj1 antibody. The number of Tuj1-positive retinal neurons was greatly increased in embryonic day 11.5 (E11.5) *Tsc1-cko* mouse retinas, expanding the neurogenic wavefront farther to the distal retina than was observed in *Tsc1-het* littermate mouse retinas (Fig. 2a). The larger numbers of Tuj1-positive cells showed stronger pS6 signals in *Tsc1-cko* mouse retinas than was observed in *Tsc1-het* mouse retinas (Fig. 2b), suggesting that cell autonomous activation of mTORC1 might accelerate retinal neurogenesis. Consistent with this, the numbers of islet-1-positive RGCs and calbindin-positive horizontal and amacrine cells in E10.5 *Tsc1-cko* mouse retinas and Crx-positive cone photoreceptors (cPRS) in E12.5 *Tsc1-cko* mouse retinas were increased, respectively, compared with their *Tsc1-het* littermate mouse retinas (Fig. 2c [left three columns], d). Not only was the production of these early-born retinal cell types more prominent in *Tsc1-cko* mouse retinas, so too was the production of late-born retinal cell types, including rhodopsin-positive rod photoreceptors, Lhx3-positive cone bipolar cells (BPS), and glutamine synthase-positive MG (Fig. 2c [right three columns], d). This completion of retinal development ahead of schedule is consistent with the results obtained in the *Pten-cko* mouse retina[14].

**Accelerated cell cycle progression in *Tsc1*-deficient RPCs.** We next examined whether RPCs in the *Tsc1-cko* mouse retina, like

those in the precociously maturing *Pten-cko* mouse retina[14], are also hyperproliferative. Both bromodeoxyuridine (BrdU)-labeled proliferating cell number and pH3-positive mitotic cell number were increased in *Tsc1-cko* mouse retinas compared with those in *Tsc1-het* littermate mouse retinas during embryogenesis (Supplementary Fig. 2a–d). However, RPC population, which can be marked by Hes1 and Vsx2, is not greatly different between the *Tsc1-het* and *Tsc1-cko* littermate mice (Supplementary Fig. 2e, f). Thus, the increased numbers of proliferating cells in those mouse retinas were not resulted from the expansion RPC population, but were an indication of the increase of RPC population in active cell cycle.

The BrdU-positive cells in the *ROSA26^lacZ* reporter (R26R), which is expressed only after Cre-mediated excision of *loxP*-STOP-loxP gene cassette knocked in *ROSA26* gene locus[25] and represents *Tsc1*-deficient cells in *Tsc1-cko;R26^lacZ/+* mouse retina, were enriched to a greater extent than those cell population in littermate *Tsc1-het;R26^lacZ/+* mouse retina (Supplementary Fig. 3c,d). The results suggest an autonomous hyperproliferation of R26R-positive, *Tsc1*-deficient RPCs resulted in faster expansion of *Tsc1*-deficient cell population than neighboring R26R-negative, *Tsc1*-positive WT RPCs (Supplementary Fig. 3a,b). This hypothesis was further supported by the results observed in P7 *Tsc1^f/f*;*TRP1-Cre;R26^EYFP/+* mouse retinas, in which RPCs affected by the tyrosinase-related protein 1 (*TRP1*)-*Cre* are derived from ciliary margin and generate retinal neurons in isolated clones[26]. Not only total numbers of R26EYFP-positive, Cre-affected cells but also the numbers of cells comprising an individual R26EYFP-positive cell clone were elevated in P7 *Tsc1^f/f*;*TRP1-Cre;R26^EYFP/+* mouse retina in comparison to those in *Tsc1^f/+*;*TRP1-Cre;R26^EYFP/+* littermate mouse retina (Supplementary Fig. 3e–h). Collectively, these results therefore account for the fact that the R26R-positive, *Tsc1*-deficient population occupies ~90% of the retinal area of P14 *Tsc1-cko;R26^lacZ/+* mice, whereas the R26R-positive, *Tsc1*-heterzygous population comprises ~75% in *Tsc1-het;R26^lacZ/+* littermate mouse retina (Supplementary Fig. 3a, b).

We next assessed the speed of RPC division in mouse retinas indirectly by counting the number of RPCs that completed first round of cell cycle after incorporating chlorodeoxy-uridine (CldU) into their chromosomes for 12 h and then proceeded to

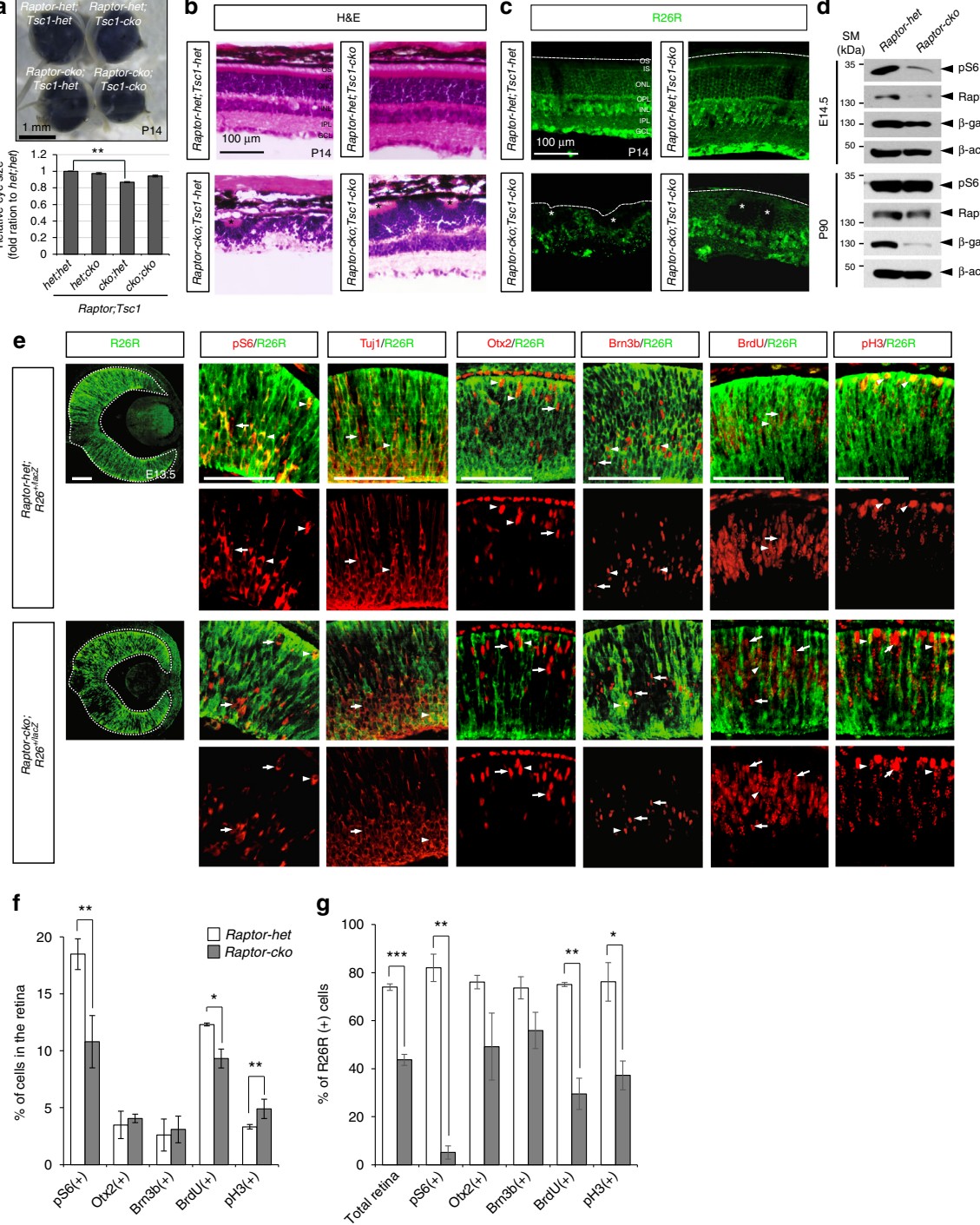

**Fig. 4** *Raptor*-deficient RPC fails to progress cell cycle. **a** Top, images of P30 *Raptor-het;Tsc1-dhet, Raptor-het;Tsc1-cko, Raptor-cko;Tsc1-het,* and *Raptor-cko; Tsc1-cko* littermate mouse eyes. Bottom, relative eye sizes are shown in a graph by comparing average diameter of the eyes. Number of eyes measured the size is 8 (*Raptor-het;Tsc1-het* and *Raptor-het;Tsc1-cKO*), 6 (*Raptor-cko;Tsc1-het*), and 5 (*Raptor-cko;Tsc1-cko*). **P-value < 0.01 (ANOVA test). **b** H&E staining images of P14 *Raptor-het;Tsc1-het, Raptor-het;Tsc1-cko, Raptor-cko;Tsc1-het,* and *Raptor-cko;Tsc1-cko* littermate mouse retinal sections. **c** Distribution of *Raptor*-deficient cells in P14 littermate mouse retina was visualized indirectly by immunodetection of R26R, which is expressed from Cre-recombined *R26^lacZ* loci. **d** Levels of pS6, Raptor, and R26R (ß-gal) in E14.5 and P90 *Raptor-het* and *Raptor-cko* littermate mouse retinas were compared by WB with corresponding antibodies. Relative amounts of protein loaded in each lane were determined by WB detection of ß-actin. **e** Distributions of retinal markers in E13.5 *Raptor-het;R26R^+/lacZ* and *Raptor-cko;R26R^+/lacZ* littermate mouse embryonic retinas were examined by immunostaining with corresponding antibodies. The R26R-positive cells (green) are potentially the *Raptor*-deficient cells. BrdU was injected to pregnant mice at 3 h prior to embryo collection. Arrows and arrowheads indicate R26R-negative and R26R-positive cells that express specific markers (red), respectively. Scale bar, 100 μm. **f** Percentage of retinal cells expressing each marker in the retinas was provided in a graph (representative images are shown in Supplementary Fig. 7). **g** Ratios of R26R co-expressing cells to total marker-expressing cells in **a** are shown in a graph. Numbers of samples used for obtaining graphs in **b** and **c** are 4 (from 3 independent litters); error bars denote SD. *P*-values are obtained by Student's *t*-test and shown in the graphs (*<0.05; **<0.01; ***<0.001)

second round to incorporating iododeoxyuridine (IdU) for 3 h. The ratio of CldU/IdU double-labeled cells, which are the population had entered into second round of cell cycle after completing first round, to total CldU-labeled proliferating cells was higher in E14.5 *Tsc1-cko* mouse retinas than in *Tsc1-het* littermate mouse retinas (Fig. 2e [rightmost column], f). To dissect the cell cycle of mouse RPCs, we also counted the number of pH3-positive mitotic cells that had entered G2 interphase and/ or mitotic (M) phases of the cell cycle within 3 h after incorporating BrdU during S phase. The number of pH3-positive cells co-labeled with BrdU was higher in E14.5 *Tsc1-cko* mouse retinas than in *Tsc1-het* littermate mouse retinas (Fig. 2e [leftmost column], f). The population of Brn3b-expressing RGCs and Otx2-positive cPRs that incorporated BrdU to their chromosomes during a 12-h period prior to exiting the cell cycle were also increased in *Tsc1-cko* mouse retinas (Fig. 2e [two center columns], f). These results suggest that daughter cells of the *Tsc1*-deficient RPCs not only progress to next round of cell cycle faster than *Tsc1-het* cells, they also exit from the cell cycle to produce more neurons during the same time period. However, over-produced retinal cells from *Tsc1-cko* mouse RPCs were eliminated by apoptotic cell death (Supplementary Fig. 4), adjusting the final number of retinal cells to a value similar to that in *Tsc1-het* mouse retinas (Fig. 1d, e).

**Rapamycin inhibits retinal neurogenesis by interfering with progenitor cell cycle progression.** We next tested whether inhibition of mTORC1 exerted effects on RPC proliferation and retinal neurogenesis opposite those of mTORC1 activation. To accomplish this, we injected pregnant WT and *Tsc1-cko* female mice with rapamycin, a chemical inhibitor of mTORC1[27, 28], at E12 days post coitum. By E12.5, pS6 signals had disappeared in embryonic retinas isolated from the rapamycin-injected mice regardless of their genotypes (Fig. 3a [leftmost column], f), indicating successful inhibition of mTORC1 by rapamycin. Rapamycin injection also significantly decreased the number of Tuj1-positive retinal neurons in those mouse embryos compared with that observed in retinas isolated from vehicle-injected WT and *Tsc1-cko* mouse embryos at the same developmental stage (Fig. 3a [second left column], c). Similar neurogenic defects were also observed in mouse embryos treated with an mTOR inhibitor, Torin1 (Supplementary Fig. 5a,b).

Inhibition of mTORC1 not only influenced retinal neurogenesis but it also affected RPC proliferation, decreasing the number of BrdU-labeled cells in WT and *Tsc1-cko* mouse embryonic retinas (Fig. 3a [second right column], c). Interestingly, the number of pH3-positive mitotic RPCs was rather increased in embryonic retinas from rapamycin-injected mice (Fig. 3a [rightmost column], c). The ratio of BrdU;pH3 double-positive cells to total pH3-positive cells, however, was not accordingly increased, but was decreased, in the rapamycin-treated E14.5 WT mouse retinas (Fig. 3b [leftmost column], d). The rapamycin treatment also decreased the BrdU;pH3 double-positive cell population in *Tsc1-cko* mouse retinas compared to that in vehicle-treated E14.5 *Tsc1-cko* mouse retinas (Fig. 3b [bottom row of leftmost column], d). The populations of CldU;IdU double-positive RPCs, BrdU; Otx2 double-positive post-mitotic photoreceptors, and BrdU; Brn3b double-positive post-mitotic RGCs were also all decreased in rapamycin-treated groups regardless of genotype (Fig. 3b [right three columns], d). These results suggest that increase of pH3 is a reflection of mitotic cell cycle arrest in the rapamycin-treated mouse retinas while decrease of BrdU represents reduced entry to S phase from G1. Collectively, these results imply that rapamycin treatment interferes with cell cycle progression of RPCs and

consequently impairs retinal neurogenesis as well as cell cycle re-entry of RPC.

**Cell cycle progression is defective in *Raptor*-deficient RPCs.** The mTORC1 complex contains an essential component Raptor, which is replaced by Rictor in mTORC2[29, 30]. To investigate whether rapamycin-induced suppression of RPC proliferation and retinal neurogenesis were specifically mediated by inhibition of mTORC1, we examined retinal development in *Raptor^flox/flox*; *Chx10-Cre* (*Raptor-cko*) mice, which lack *Raptor* and lose mTORC1 activity in RPCs and descendent retinal neurons. The average size of *Raptor-cko* mouse eyes was about 87% that of *Raptor-het* littermate eyes; their retinas were also thinner than normal and showed multiple rosette structures (Fig. 4a, b). Interestingly, R26R-positive cells, which are *Raptor*-deficient, were hardly detectable in mature *Raptor-cko* mouse retinas (Fig. 4c, d), but were present in significant numbers in the embryonic stages (Fig. 4d, e [leftmost column]). However, there were already significantly fewer R26R-positive cells in *Raptor-cko* embryonic mouse retinas than in *Raptor-het* littermate mouse retinas (Supplementary Fig. 6d). Not only total apoptotic cell numbers but also R26R-positive apoptotic cell population were significantly increased in the *Raptor-cko* embryonic mouse retinas (Supplementary Fig. 6a, b, c, e, f), implicating preferential loss of *Raptor*-deficient cells by cell death. The results suggest that *Raptor*-deficient cells are eliminated by apoptosis in the embryonic mouse retinas, resulting in the depletion of R26R-positive *Raptor*-deficient cell population in the post-natal *Raptor-cko* mouse retinas.

In contrast to the lack of pS6 signal in rapamycin-treated retinas, pS6-positive cells were still present at reduced number in E13.5 *Raptor-cko* mouse retinas (Fig. 4e; Supplementary Fig. 7). In addition, the numbers of neurons positive for Tuj1, Otx2, or Brn3b in the retina were not changed to as great an extent as was the case in rapamycin-treated retinas (Fig. 4e, f; Supplementary Fig. 7). However, a comparison of pS6 and R26R positivity in *Raptor-cko* mouse retinas revealed that pS6 signals were selectively enriched in R26R-negative WT retinal cells (Fig. 4e, g). A majority of cells expressing those neuronal markers in *Raptor-cko* mouse retinas were also negative for R26R (Fig. 4e [three center columns], g), suggesting that the loss of *Raptor* and consequent inactivation of mTORC1 interfered with retinal neurogenesis. The R26R-positive cells also showed a reduced ability to incorporate BrdU and exhibited diminished pH3 expression (Fig. 4e, g), indicating a proliferation defect in *Raptor*-deficient cells. In support of complementary expansion of WT cells in the retina, the numbers of mitotic RPCs (pH3-positive) and retinal neurons (Brn3b-positive and Otx2-positive), which incorporated BrdU for 3 and 12 h, respectively, were higher in the R26R-negative WT cell population than in the R26R-positive *Raptor*-deficient cell population in the E13.5 *Raptor-cko* mouse retina (Fig. 4g). Collectively, these results suggest that inactivation of mTORC1, either by rapamycin treatment or *Raptor* deletion, inhibits RPC cell cycle progression for neurogenesis as well as self-renewal, whereas activation of mTORC1 promotes those.

**Cyclin turnover in RPCs is sensitive to mTORC1 activity.** Mitotic cell cycle progression requires accumulation of cyclin B (CcnB) and consequent activation of cyclin-dependent kinase 1 (Cdk1), which promotes mitosis by phosphorylating various target proteins in G2/M phases of the cell cycle[31–33]. In the same manner, CcnE and CcnA trigger the progression to S and G2 phases from G1 and S phases, respectively, by activating Cdk2[33, 34]. Consistent with a supportive role of mTORC1 in

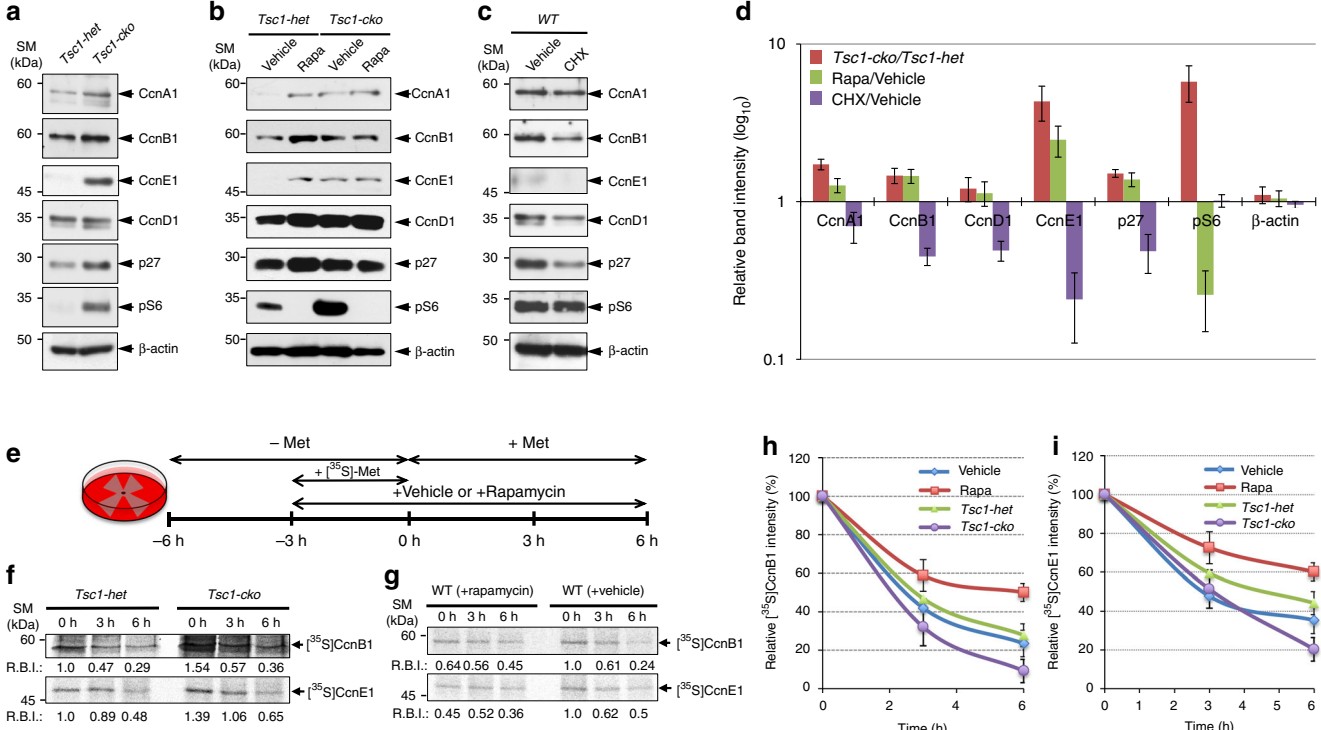

**Fig. 5** Cyclin turnover in RPC is sensitive to mTORC1 activity. **a** Proteins (20 μg) of cell lysates of E14.5 *Tsc1-het* and *Tsc1-cko* littermate mouse retinas were analyzed by WB with corresponding antibodies. Alternatively, pregnant mice can produce either *Tsc1-het* or *Tsc1-cko* mice were injected with vehicles (control in **b** and **c**), rapamycin (5 mg/kg; **b**), or cycloheximide (CHX; 5 mg/kg; **c**) at 14 dpc prior to collecting the embryonic retina at 14.5 dpc for WB analyses. **d** Image pixels of WB bands were calculated by the ImageJ software and relative intensities were obtained by comparing the pixel numbers with those of control samples (*Tsc1-het* retina for *Tsc1-cko*; vehicle-injected retinas for rapamycin and CHX). Values are average measurements of 3 independent WB results. Error bars denote SD. **e** Experimental schedule for labeling and chasing of newly synthesized cyclin proteins in mouse retinal explants. E13.5 littermate retinal explants were incubated in methionine (Met)-free growth media for 3 h and then added with [$^{35}$S]-Met (250 μCi/ml) for 3 h to label newly synthesized proteins in the retinas. The explants were then incubated in normal growth media containing unlabeled Met for the indicated time. For the analysis in **g**, WT C57BL6/J mouse retinal explants were added with growth media containing vehicle or rapamycin as indicated. **f**, **g** CcnB1 and CcnE1 in the retinal explants were isolated by immunoprecipitation with corresponding antibodies and the immunoprecipitates were analyzed by 10% SDS-PAGE followed by detection of radioactivities of [$^{35}$S]-labeled protein bands using BAS-7000 (Fuji Inc.). Relative band intensities (RBI) of [$^{35}$S]-CcnB1 and [$^{35}$S]-CcnB1 were obtained by comparing the image pixel numbers [$^{35}$S]-CcnB1 and [$^{35}$S]-CcnB1 bands of each sample with *Tsc1-het* (**f**) and vehicle-treated (**g**) samples at time 0 (0 h) using ImageJ software, and presented below the band images. To examine the decay rates of [$^{35}$S]-labeled CcnB1 (**h**) and CcnE1 (**i**) in *Tsc1-cko* and rapamycin-treated mouse retinas, relative intensities against the values at 0 h were obtained and shown in the graphs. The values are average band intensity obtained from 4 independent experiments. Error bars denote SD

mitotic cell cycle progression[27, 35], the level of CcnB1 in E14.5 *Tsc1-cko* mouse retinas was higher than that in *Tsc1-het* littermate mouse retinas (Fig. 5a, d; Supplementary Fig. 8). The levels of CcnE1 and CcnA1 were also commonly increased in the E14.5 *Tsc1-cko* mouse retinas, implicating enhanced G1-to-S and S-to-G2 cell cycle progression of mTORC1-hyperactive *Tsc1*-deficient RPCs. However, the levels of CcnD1, which is expressed in RPCs[36], were not significantly different between E14.5 *Tsc1-het* and *Tsc1-cko* mouse retinas (Fig. 5a, d), in consistent with indifferent RPC population in those littermate mouse retinas (Supplementary Fig. 2e, f).

Unexpectedly, rapamycin treatment, which interfered with cell cycle progression in mouse embryonic RPCs (Fig. 3b, d), did not decrease cyclin levels in E14.5 mouse retinas regardless of genotypes (Fig. 5b, d). It rather increased the levels of cyclin proteins in mouse retinas (Fig. 5b, d), without correspondingly increasing cyclin mRNA levels (Supplementary Fig. 8). Given the negative effects of rapamycin on translation[37], the results were quite surprising. We thus hypothesized that rapamycin treatment might accumulate cyclin proteins by interfering with the synthesis of the proteins that inhibit cyclin synthesis or promote cyclin decay. However, in opposite to the effects of rapamycin, the levels

of cyclin proteins were commonly decreased in embryonic retinas isolated from mice injected with a general translation inhibitor cycloheximide (CHX; Fig. 5c, d). These results therefore imply that cyclin accumulation in rapamycin-treated mouse retina was not resulted from the inhibitory effects of rapamycin on translation.

Cell cycle progression is not only triggered by the synthesis of cell cycle phase-specific cyclins, it is also coupled to the degradation of cyclins that had acted in previous cell cycle phases[34]. Accordingly, mTORC1 might promote RPC cycle progression by regulating cyclin protein degradation as well as synthesis. Hence, mTORC1 inactivation by rapamycin possibly resulted in cyclin accumulation through inhibition of degradation of the proteins. To test this hypothesis, we examined the rates of cyclin protein degradation in mouse embryonic retinal explants by pulse labeling newly synthesized cellular proteins with [$^{35}$S]-methionine and subsequently chasing with unlabeled methionine (Fig. 5e). The relative amounts of [$^{35}$S]-labeled CcnB1 and CcnE1 synthesized over a 3-h labeling period were higher in *Tsc1-cko* retinal explants than in *Tsc1-het* littermate retinal explants (Fig. 5f, see relative band intensity (RBI) values), reaffirming the translation-stimulating activity of mTORC1[37]. However, the

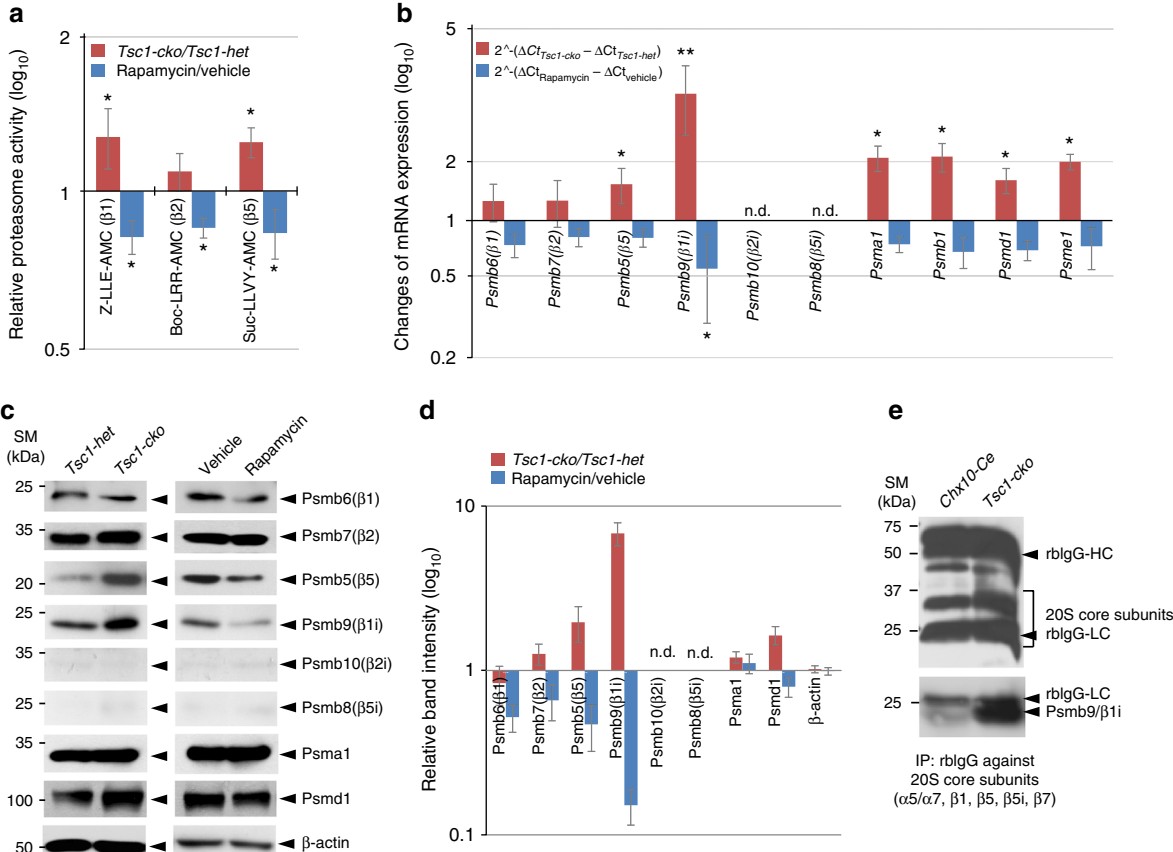

**Fig. 6** mTORC1-induced developmental acceleration of the retina requires an immunoproteasome subunit Psmb9. **a** The 20S proteasomes were purified from E14.5 *Tsc1-het* and *Tsc1-cko* littermate mouse retinas by 20S Proteasome Purification Kit® (Enzo Life Sciences), and caspase-like (β1/β1i), trypsin-like (β2/β2i), and chymotrypsin-like (β5/β5i) activities of the purified proteasomes were analyzed by measuring fluorescent intensities produced by cleaved peptide substrates Z-LLE-AMC (β1/β1i), Bz-VGF-AMC (β2/β2i), and Scu-LLVY-AMC (β5/β5i) (see details in Methods). The values are averages and error bars denote SD ($n = 3$; 3 independent litters). **b** Relative mRNA levels of 20S proteasome subunits in E14.5 *Tsc1-cko* and rapamycin-treated mouse retinas were obtained by comparing real-time quantitative PCR (RT-qPCR) values with those of *Tsc1-het* and vehicle-treated mouse retinas at the same age. The values are averages obtained by 6 (*Tsc1-cko/Tsc1-het*) and 7 (rapamycin/vehicle) independent measurements with mRNA isolated from 4 (*Tsc1-cko/Tsc1-het*) and 6 (rapamycin/vehicle) independent batches. **c** Relative levels of 20S proteasome subunit proteins in the mouse retinas were also examined by WB with corresponding antibodies. **d** Image pixels of WB bands in *Tsc1-cko* samples were calculated by the ImageJ software and relative intensities were obtained by comparing the pixel numbers with those *Tsc1-het* samples. Values are average measurements of 4 independent WB results. **e** Incorporation of Psmb9 into the proteasome was examined by WB detection of Psmb9 in 20S proteasome core complex, which was isolated by the 20S Proteasome Purification Kit. *P*-values are obtained by Student's *t*-test and shown in the graphs (\*<0.05; \*\*<0.01)

levels after a 3-h chase period were not greatly different between the two genotypes, and were further decreased in *Tsc1-cko* retinal explants after a 6-h chase (Fig. 5h, i). These results suggest that [35S]-labeled cyclins persist for a shorter period of time in *Tsc1-cko* mouse retinas than in *Tsc1-het* littermate mouse retinas.

Conversely, the relative amounts of [35S]-labeled cyclins in retinal explants treated with rapamycin during a 3-h labeling period were lower than those in vehicle-treated retinal explants (Fig. 5g, see RBI values). The decay of [35S]-labeled cyclins was also slower in the rapamycin-treated retinal explants. About 50% of the [35S]-labeled CcnB1 was still detectable in rapamycin-treated retinas 6 h post incubation, whereas only 23% of [35S]-labeled CcnB1 remained in vehicle-treated samples at the same time point (Fig. 5h). A similar decay pattern was also observed from [35S]-labeled CcnE1, suggesting that the turnover rate of CcnE1 is also sensitive to mTORC1 activity (Fig. 5i). However, the total amount of [35S]-labeled protein in the retinas was not sensitive to mTORC1 activity (Supplementary Fig. 9a, b), suggesting that mTORC1-regulated protein decay is not applicable to all proteins, but might be specific for certain proteins, including CcnB1 and CcnE1. Taken together, these results suggest that mTORC1 not only facilitates the synthesis of cyclins, it also promotes the degradation of these proteins. The short oscillatory cycle of cyclin synthesis and degradation might drive mTORC1-activated RPCs to progress through the cell cycle faster than control RPCs.

**Elevated proteasomal activity in the *Tsc1-cko* mouse retina.** The decay of cyclins during the cell cycle is mainly mediated by ubiquitin-dependent proteasomal degradation[34], suggesting that enhanced cyclin turnover in the *Tsc1-cko* mouse retina might result from enhanced ubiquitination of cyclins. The relative level of ubiquitinated CcnB1 was, however, lower in E14.5 *Tsc1-cko* mouse retinas than in littermate *Tsc1-het* mouse retinas—the inverse of the higher CcnB1 level in *Tsc1-cko* mouse retinas (Supplementary Fig. 10a, b). On the contrary, the levels of other ubiquitinated proteins were not significantly different between the *Tsc1-het* and *Tsc1-cko* littermate mouse retinas (Supplementary Fig. 10f). Ubiquitinated CcnB1 levels became

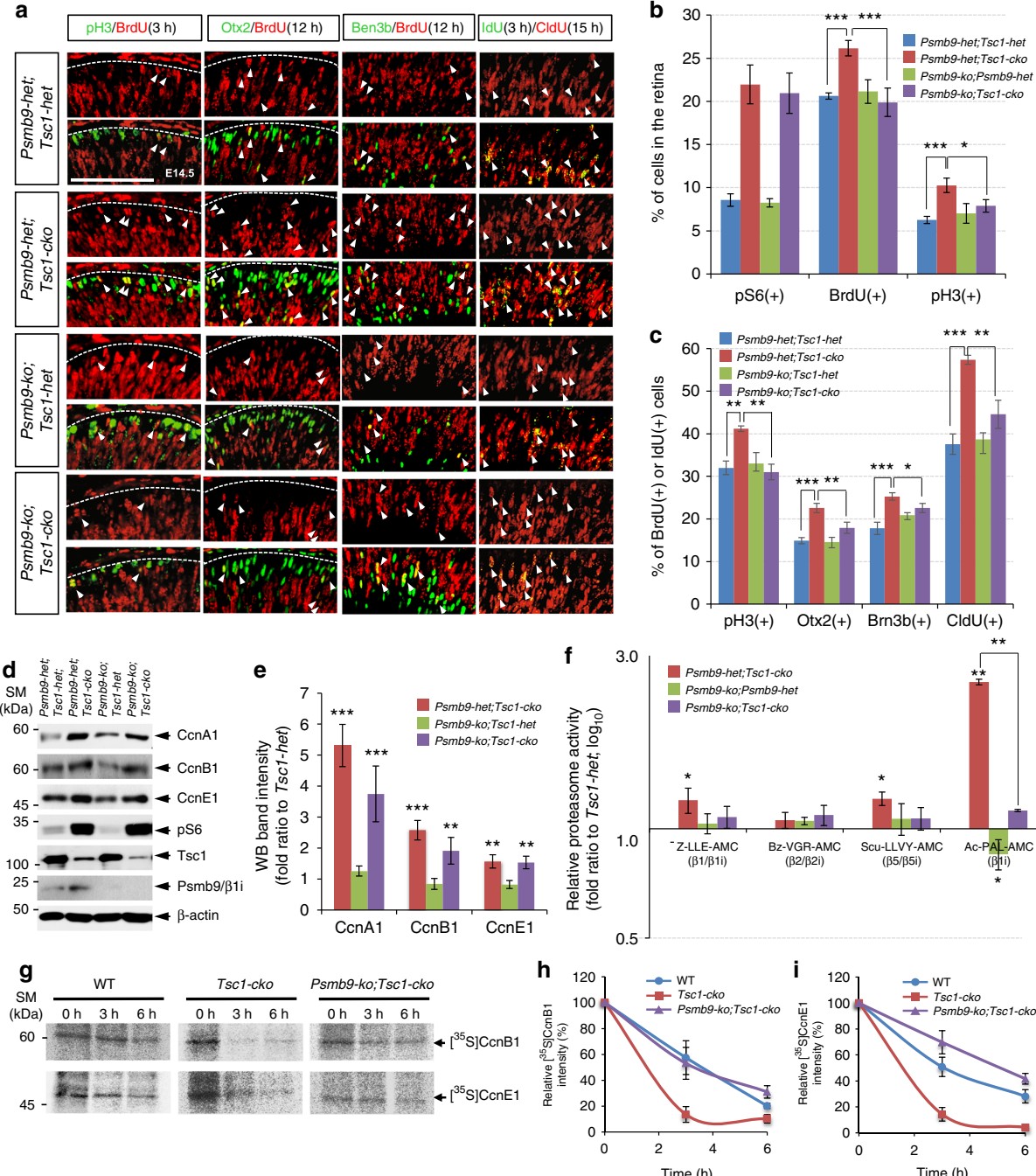

**Fig. 7** Psmb9 is necessary for the accelerated cyclin turnover in *Tsc1-cko* RPC. **a** Cell cycle progression of RPCs in E14.5 mouse retinas with combinatorial *Tsc1-cko* and *Psmb9-ko* alleles was examined as described in Fig. 2f. Scale bar, 100 μm. **b** Average percentage of cells expressing each marker in the retinas was provided in a graph (representative images are shown in Supplementary Fig. 12a). **c** Ratio of BrdU-co-expressing cells to total marker-expressing cells in (**a**) was also shown in a graph. Numbers of mice used for **b** and **c** are 4 (*Tsc1-cko;Psmb9-ko*) or 5 (the rest genotypes) and from 4 independent litters; error bars denote SD. **d** Retinal cell lysates (20 μg proteins/lane) obtained from E14.5 littermate mouse embryos were analyzed by WB with corresponding antibodies. **e** Image pixels of WB bands in each retinal sample were calculated by ImageJ software and relative intensities were obtained by comparing the pixel numbers with those of *Psmb9-het;Tsc1-het* samples. Values are average measurements of 3 independent WB results. Error bars denote SD. **f** Activities of the purified proteasomes were analyzed as described in Fig. 6a. The values are averages and error bars denote SD (*n* = 4; 2 independent litters). **g** Synthesis and degradation of CcnB1 in the mouse retinas were analyzed by [$^{35}$S]-Met pulse labeling and chasing experiments explained in Fig. 5e. Relative radioactivities of [$^{35}$S]-labeled CcnB1 (**h**) and CcnE1 (**i**) bands in the retinal explants were obtained by comparing corresponding bands at 0 h. The values are average obtained from 3 independent experiments. Error bars denote SD. *P*-values are obtained by one-way ANOVA test and shown in the graphs (*<0.05; **<0.01; ***<0.001)

indistinguishable between E14.5 *Tsc1-het* and *Tsc1-cko* littermate mouse retinas after the inhibition of the proteasomes by MG-132. The results implicate that the amounts of ubiquitinated CcnB1 gained after MG-132 treatment were higher in *Tsc1-cko* mouse retinas in comparison to those in *Tsc1-het* littermates (Supplementary Fig. 10a, b, e). Together, the results suggest that degradation of ubiquitinated CcnB1 is more efficient in the *Tsc1-cko* mouse retinas than that in the *Tsc1-het* mouse retinas.

Conversely, the levels of ubiquitinated CcnB1 in mouse embryonic retinas were increased by rapamycin treatment (Supplementary Fig. 10c). MG-132 treatment did not cause further accumulation of ubiquitinated proteins in rapamycin-injected mouse retinas, suggesting that proteasomal degradation of ubiquitinated CcnB1 was already inhibited in the rapamycin-treated retinas (Supplementary Fig. 10c–e). Collectively, these results suggest that proteasome-mediated degradation of ubiquitinated CcnB1, but not the rate of CcnB1 ubiquitination, is sensitive to mTORC1 activity.

In support of this, the activities of caspase-like (responsible by β1 subunit) and chymotrypsin-like proteases (responsible by β5 subunit), which constitute the catalytic subunits of the 20S proteasome core particle together with trypsin-like (responsible by β2 subunit) proteases[38], were ~1.2-fold higher in E14.5 *Tsc1-cko* mouse retinas than in *Tsc1-het* littermate mouse retinas (Fig. 6a). Conversely, rapamycin treatment decreased the proteasome activities in E14.5 mouse retinas to about 80% of that observed in vehicle-treated retinas (Fig. 6a). Taken together with the [35S]-Met pulse-chase results (Fig. 5f), these findings suggest that elevated proteasome activity might be responsible for the accelerated degradation of CcnB1 in the *Tsc1-cko* mouse retina.

**mTORC1 stimulates proteasome gene expression in the mouse retina**. It was reported that expression of 20S proteasome core particle subunit genes is enhanced in *Tsc2-ko* mouse embryonic fibroblasts and adult mouse brain neurons, in which mTORC1 activity is also increased[39]. Thus, to determine whether this mTORC1-induced proteasome gene expression might account for the increased proteasome activity in *Tsc1-cko* retinas, we examined the levels of proteasome subunit mRNAs and proteins in E14.5 *Tsc1-het* and *Tsc1-cko* mouse retinas. The relative mRNA levels of 20S constitutive proteasome catalytic subunits β1 (*Psmb6*), β2 (*Psmb7*), and β5 (*Psmb5*) were changed insignificantly or increased <1.5-fold in E14.5 *Tsc1-cko* mouse retinas compared with *Tsc1-het* littermate mouse retinas (Fig. 6b). The mRNA levels of these proteasome subunits were decreased mildly (to ~80% levels of untreated samples) by rapamycin treatment (Fig. 6b). The levels of non-catalytic subunits of proteasomes, including the 20S subunits Psma1 and Psmb1, the 19S regulatory particle subunit Psmd1, and the 11S regulatory particle subunit Psme1, were changed between 1.5- and 2-fold (Fig. 6b), suggesting increase and decrease in proteasome numbers in *Tsc1-cko* and rapamycin-treated mouse retinas, respectively. However, corresponding changes in protein levels in *Tsc1-cko* and rapamycin-treated mouse retinas compared with control *Tsc1-het* and vehicle-treated samples were difficult to detect, except for β5 (Psmb5) and Psmd1 (Fig. 6c, d; Psmb1 and Psme1 protein levels were not examined).

It was shown that a precursor form of immunoproteasome subunits interacts with mTORC1 via Pras40 in prior to its maturation and incorporation into a functional immunoproteasome[40]. Interestingly, mRNA and protein levels of Psmb9/β1i, an inducible β1 isoform in the immunoproteasome[41], showed the greatest increase in *Tsc1-cko* mouse retinas and decrease in rapamycin-treated mouse retinas (Fig. 6b–d; Supplementary Fig. 11a). Furthermore, the Psmb9 was also successfully incorporated into 20S proteasomes in the *Tsc1-cko* mouse retina (Fig. 6e), suggesting the formation of functional Psmb9-containing immunoproteasomes in the retina. These results suggest that immunoproteasome catalytic subunits might be one that contributes significantly to elevate cellular proteasome activity in *Tsc1-cko* mouse retinas, where Psmb6/β1 and Psmb7/β2 were not increased significantly (Fig. 6c, d). However,

the other immunoproteasome subunits, Psmb8/β5i and Psmb10/β2i, were not expressed at significant levels in mouse retinas, although they were increased in E17.5 *Tsc1^f/f^;Emx1-Cre* mouse cerebrum and decreased in the rapamycin-treated adult mouse spleen (Fig. 6b–d; Supplementary Fig. 11b).

**Psmb9 is necessary for the developmental acceleration of *Tsc1-cko* mouse retinas**. To delineate the roles of Psmb9 in mTORC1-promoted retinal development, we generated combinatorial *Tsc1-cko* and *Psmb9-ko* mice. Deletion of *Psmb9* alone did not cause remarkable anatomical changes in embryonic or adult mouse retinas (Supplementary Fig. 12a; results of adult retina are not shown). The numbers of proliferating RPCs and retinal neurons in E14.5 *Psmb9-ko* mouse retinas were also not greatly different from those in WT and *Psmb9-het* littermates (Fig. 7a, b; Supplementary Fig. 12a; WT results are not shown). In contrast, the combined loss of *Psmb9* and *Tsc1* in E14.5 *Psmb9-ko;Tsc1-cko* mouse retinas significantly decreased the number of Tuj1-positive retinal neurons compared with that observed in *Psmb9-het;Tsc1-cko* littermate mouse retinas (Supplementary Fig. 12a, second row). There were also reduced numbers of BrdU-positive proliferating RPCs and pH3-positive mitotic RPCs in *Psmb9-ko;Tsc1-cko* mouse retinas compared with those in *Psmb9-het;Tsc1-cko* littermate mouse retinas (Fig. 7b; Supplementary Fig. 12a, third and fourth rows). The populations of BrdU;pH3, BrdU;Otx2, BrdU;Brn3b, and CldU;IdU double-positive cells in *Psmb9-ko;Tsc1-cko* mouse retinas were also all smaller than those in *Psmb9-het;Tsc1-cko* littermate mouse retinas and indistinguishable from those in *Psmb9-het;Tsc1-het* and *Psmb9-ko;Tsc1-het* mouse retinas (Fig. 7a, c), suggesting that Psmb9 is necessary for cell cycle acceleration in *Tsc1*-deficient RPCs. However, the number of pS6-positive retinal cells in *Psmb9-ko;Tsc1-cko* mouse retinas was not different from that in littermate *Psmb9-het;Tsc1-cko* mouse retinas (Fig. 7b; Supplementary Fig. 12a), reinforcing the conclusion that Psmb9 acts downstream of or in parallel with mTORC1 activation.

The effects of *Psmb9* gene loss on the proliferation of *Tsc1*-deficient RPCs were also tested in vitro. The number of cells in a neurosphere derived from a *Tsc1*-deficient RPC is larger than that from a *Tsc1-het* RPC, implicating that *Tsc1*-deficient RPCs expanded faster than *Tsc1-het* RPCs during the culture (Supplementary Fig. 12b, c). Not only total number of cells but also the number of BrdU;pH3 double-positive cell population in a *Tsc1-cko;Psmb9-ko* retinal neurosphere were decreased in comparison to *Tsc1-cko* retinal neurospheres (Supplementary Fig. 12b–d), suggesting that the loss of *Psmb9* inhibits the expansion of *Tsc1*-deficient RPC clone by decelerating cell cycle progression.

Despite the deceleration of cell cycle progression of *Tsc1-cko* mouse RPCs by concomitant loss of *Psmb9*, the levels of CcnA1, CcnB1, and CcnE1 observed in *Psmb9-ko;Tsc1-cko* mouse retinas were not fully restored to the levels observed in *Psmb9-het;Tsc1-het* littermate mouse retinas (Fig. 7d, e). This might be related with the reduced proteasomal activity, especially the immunoproteasome activity, in the retinas. In support of this, the activity of immunoproteasomes, determined by assessing cleavage of the specific substrate Ac-PAL-AMC, was significantly increased in E14.5 *Tsc1-cko* mouse retinas compared with that in *Psmb9-het;Tsc1-het* littermate mouse retinas, but was almost undetectable in *Psmb9-ko;Tsc1-cko* littermate mouse retinas (Fig. 7f). Furthermore, the activity of the caspase-like proteasome β1/β1i subunit was also decreased significantly in *Psmb9-ko;Tsc1-cko* mouse retinas relative to *Psmb9-het;Tsc1-cko* littermate mouse retinas, suggesting that Psmb9 is mainly responsible for the increase in caspase-like proteasome activity in the *Tsc1-cko* mouse retina.

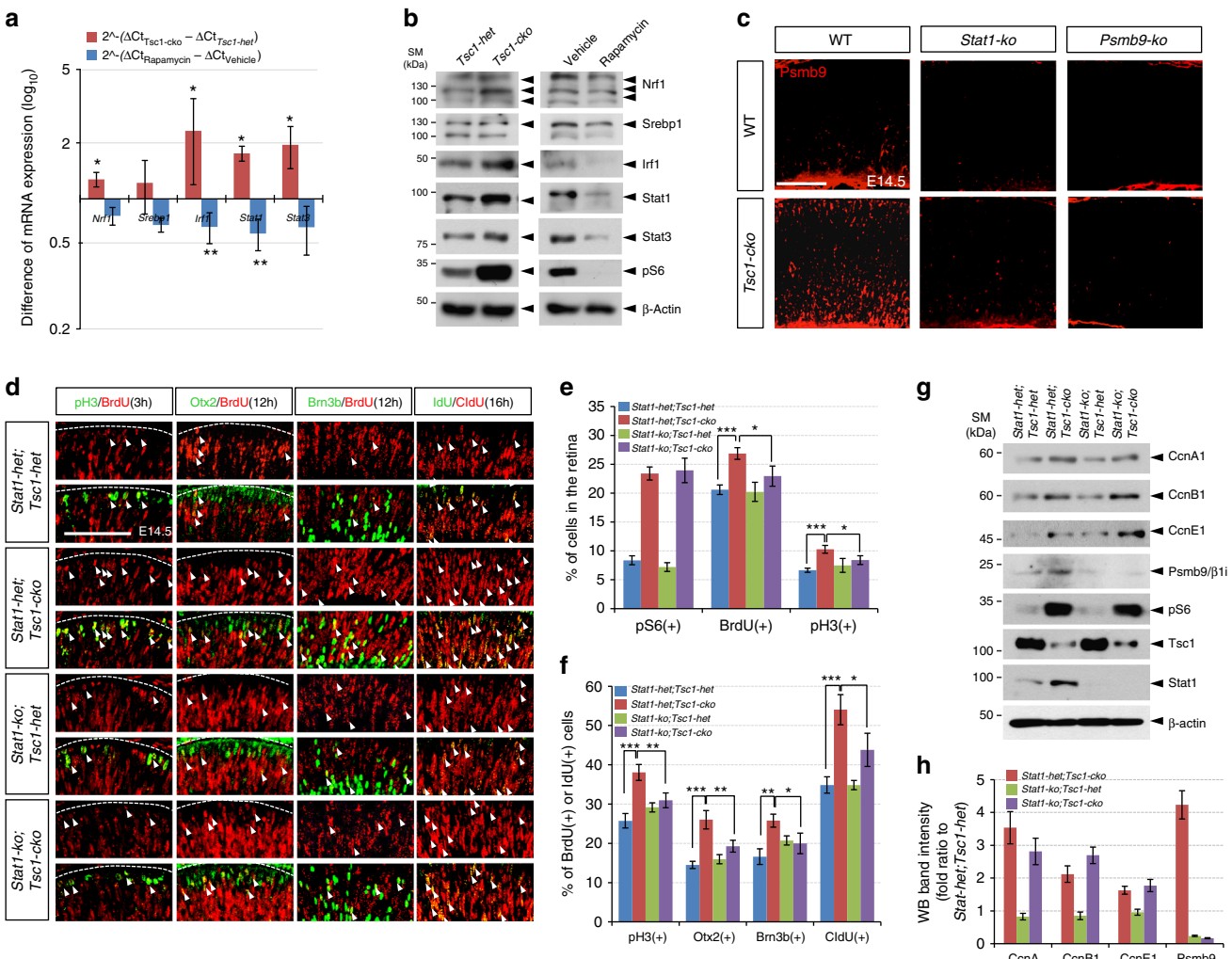

**Fig. 8** mTORC1 mediates Stat1 to induce Psmb9 in RPCs. **a** Relative mRNA levels of transcription factors, which have been known as regulators of proteasome subunit expression[39, 42, 46], in E14.5 *Tsc1-cko* and rapamycin-treated mouse retinas were obtained by comparing RT-qPCR values with those of *Tsc1-het* and vehicle-treated mouse retinas at the same age. Error bars denote SD (*n* = 5). **b** The transcription factor proteins in E14.5 mouse retinas were analyzed by WB with corresponding antibodies. Multiple bands of Nrf1 are products of glycosylation[62]. **c** Distribution of Psmb9 in E14.5 mouse retinas with indicated genotypes was examined by immunostaining with anti-Psmb9 antibody. Scale bar, 100 μm. **d** Cell cycle progression of RPCs in E14.5 mouse retinas with combinatorial *Tsc1-cko* and *Psmb9-ko* alleles was examined as described in Fig. 2f. Scale bar, 100 μm. **e** Average percentage of cells expressing each marker in the retinas was provided in a graph. **f** Ratio of BrdU-co-expressing cells to total marker-expressing cells in **d** was also shown in a graph. Numbers of mice used are 4 and from 3 independent litters; error bar denotes SD. **g** Retinal cell lysates (20 μg proteins/lane) obtained from E14.5 littermate mouse embryos were analyzed by WB with corresponding antibodies. **h** Image pixels of WB bands in each retinal sample were calculated by ImageJ software and relative intensities were obtained by comparing the pixel numbers with those of *Stat1-het;Tsc1-het* samples. Values are average measurements of 3 independent WB results. Error bars denote SD. *P*-values are obtained by one-way ANOVA test and shown in the graphs (*<0.05; **<0.01; ***<0.001)

These results implicate that cyclins in *Psmb9-ko;Tsc1-cko* mouse retinas are synthesized as fast as those in *Tsc1-cko* mouse retinas, however their proteasomal degradation is less efficient.

To test this hypothesis, we compared the degradation of [35S]-labeled CcnB1 in E14.5 *Psmb9-ko;Tsc1-cko* mouse retinal explants with those in E14.5 WT and *Tsc1-cko* mouse retinal explants. These experiments showed that the decay of [35S]-labeled CcnB1 and CcnE1 in *Psmb9-ko;Tsc1-cko* mouse retinal explants was slower than that in *Tsc1-cko* mouse retinal explants (Fig. 7g–i; Supplementary Fig. 9c). Collectively, these results suggest that activated mTORC1 in *Tsc1*-deficient mouse RPCs induces Psmb9 and consequently the formation of Psmb9-containing immunoproteasomes, which in turn degrade accumulated cellular substrates, including CcnB1 and CcnE1, to accelerate cell cycle progression.

**mTORC1 acts through Stat1 to induce *Psmb9* in mouse RPCs.**
Psmb9 expression is known to be regulated by various transcription factors, including interferon regulatory factor 1 (Irf1), Stat1, and nuclear factor-κB (NF-κB), in response to interferon-γ (IFN-γ) and tumor necrosis factor-α[42, 43]. The level of inhibitor of NF-κB, which is a transcription target of NF-κB that binds to NF-κB and interferes with its nuclear translocation[44], was not substantially different between *Tsc1-het* and *Tsc1-cko* mouse retinas (data not shown). On the contrary, both mRNA and protein levels of Stat1 and Irf1 in E14.5 *Tsc1-cko* mouse retinas were increased significantly compared with those in littermate *Tsc1-het* mouse retinas (Fig. 8a, b). Stat1 has been shown to interact with mTOR[45] and upregulate *Psmb9* gene expression directly or indirectly via *Irf1* induction[46]. These results therefore suggest that

the Stat1 could be a candidate for connecting mTORC1 activation to Psmb9 induction.

To test this hypothesis, we created *Stat1-ko;Tsc1-cko* mice in which both *Stat1* and *Tsc1* were eliminated. Psmb9 expression in E14.5 *Stat1-ko;Tsc1-cko* mouse retinas was almost undetectable (Fig. 8c, g). Similar to the results obtained in *Psmb9-ko* mouse retinas, deletion of *Stat1* alone did not result in significant changes in mouse retinas (Fig. 8d [third row]–f). In contrast, not only the numbers of CldU;IdU double-labeled cells but also those of mitotic RPCs (pH3-positive) and PMNs (Brn3b-positive and Otx2-positive), which were co-labeled with BrdU for 3 and 12 h, respectively, were decreased significantly in E14.5 *Stat1-ko;Tsc1-cko* mouse retinas compared with *Stat1-het;Tsc1-cko* littermate mouse retinas (Fig. 8d, f), suggesting that the absence of Stat1 normalizes the cell cycle speed in *Tsc1*-deficient RPCs.

The levels of CcnA1, CcnB1, and CcnE1 in *Stat1-ko;Tsc1-cko* mouse retinas were not fully recovered to normal levels (Fig. 8g, h), similar to the results obtained in *Psmb9-ko;Tsc1-cko* mouse retinas (Fig. 7e). The recovery of regular cell cycle period without the normalization of cyclin protein levels in *Psmb9-ko;Tsc1-cko* and *Stat1-ko;Tsc1-cko* mouse retinas might be accounted by the delayed degradation of cyclin proteins, which were synthesized rapidly by elevated mTORC1 activity, in the absence of Psmb9 induction. Collectively, these results suggest that the Stat1-(Irf1)-Psmb9 cascade plays an important role in accelerating cell cycle to rapidly expand the *Tsc1*-deficient retinal cell population and develop the *Tsc1-cko* mouse retina earlier than dictated by the regular developmental schedule.

## Discussion

Our study demonstrates a role for mTORC1 in coordinately regulating the synthesis and degradation of proteins in mouse RPCs. The latter effects of mTORC1 are mediated by upregulation of proteasome subunits, especially the immunoproteasome subunit Psmb9 in the RPCs. The immunoproteasome subunits were also elevated in the *Tsc1*<sup>*flox/flox*</sup>;*Emx1-Cre* mouse embryonic brain and were decreased in the rapamycin-treated adult mouse spleen (Supplementary Fig. 11), suggesting that induction of the immunoproteasome is a common outcome of mTORC1 activation. By elevating the levels of proteasomes—both constitutive proteasomes and immunoproteasomes (Fig. 6b–d)—mTORC1 might be able to terminate cellular events that are induced by the proteins subjected to proteasomal degradation. However, the degradation of total cellular proteins in *Tsc1-cko* mouse retinas and rapamycin-treated mouse retinas are not accordingly changed (Supplementary Fig. 10), implicating the mTORC1-sensitive proteasomal degradation is not applied to all proteins but specific targets. We demonstrate CcnB1 and CcnE1 as examples of the proteins that are subject to mTORC1-regulated proteasomal degradation in the developing mouse retina. The cyclins were also synthesized more rapidly in *Tsc1-cko* mouse retinas potentially through translational activation (Fig. 5f). Therefore, through mTORC1-dependent coordinated synthesis and degradation of cyclin proteins, RPCs might be able to divide without interruptions caused by an insufficient supply or persistent accumulation of cyclins.

An important cellular application of the inducible feature of immunoproteasome is clearing aberrantly accumulated proteins (i.e., antigens) in antigen-presenting cells[47]. The immunoproteasome might also be useful in other cell types in degrading endogenous proteins that accumulate very rapidly and cannot be cleared by constitutive proteasomes alone. Because mTORC1 activity robustly enhances cellular protein synthesis through activation of S6 and eukaryotic translation initiation factor 4E, it would be incumbent upon cells in which mTORC1 is hyperactive

to have a mechanism capable of handling rapidly accumulated proteins. Otherwise, the cell's fate might be altered by these excessive proteins, which could persistently activate their associated cellular events or induce protein stress responses by forming aggregates[48]. Thus, induction of the immunoproteasome could also be an effective strategy for *Tsc1*-deficient cells to degrade proteins that had rapidly accumulated in the cell.

It should be noted that the role of mTORC1 in proteasome-mediated protein degradation is still a matter of debate. Zhang et al. reported that mTORC1 coordinately activates total protein synthesis and degradation. The enhanced protein degradation in *Tsc2-ko* mouse embryonic fibroblasts and neurons is mediated by Srebp1-Nrf1-dependent upregulation of proteasome subunit gene expression. However, Zhao et al. reported that mTORC1 inhibits proteasomal degradation of a majority of cellular proteins. An inhibitory role of TOR in proteasome was also proposed in another report that shows TOR post-transcriptionally down-regulates chaperones involved in the assembly of 19S regulatory particles in yeast and mammals[49]. The 20S core particles of immunoproteasomes are capable of forming mature complexes both 19S and 11S regulatory particles, whereas those of constitutive proteasome associate only with 19S[50]. Therefore, the immunoproteasomes comprised of 20S–11S particles might play important roles for protein degradation in the conditions that mTOR inhibits the assembly of 19S subunits and consequently decreases the content of mature constitutive proteasomes. Additionally, one caveat of the paper of Zhao et al. is that the inhibitory effects of mTORC1 on proteasome-mediated protein degradation were most applicable to proteins that turn over slowly, not to those with rapid turnover. In support of this idea, the levels of CcnA1, CcnB1, and CcnE1, which among cyclins have relatively short half-lives, were more sensitivity to mTORC1 activity than CcnD1, which has a longer half-life and showed no remarkable differences in levels in response to mTORC1 activity (Fig. 5a–c). By accelerating the turnover of these short-lasting cyclins in RPCs, mTORC1 would be able to shorten the time required to progress to next cell cycle phases and thus the time necessary for cell division (Supplementary Fig. 13).

As we showed in the *Tsc1-cko* mouse retina, the accumulation of cyclin A, B, and E has also been reported in the *Drosophila tsc1* mutant eye disc, and has been proposed as a cause of the hyperproliferation of *tsc1* mutant eye disc cells[51, 52]. In contrast, similar to *Raptor-cko* mouse retinas (Fig. 4; Supplementary Fig. 7), *raptor* mutant fly eyes show diminished proliferation and reduced size[53], suggesting an evolutionarily conserved role for TORC1 in retinal growth and development by regulating cell proliferation. Therefore, dTORC1-regulated retinal development in *Drosophila* might be also related to the accelerated turnover of cyclin proteins by proteasomes in addition to their facilitated synthesis. This could be accomplished by constitutive proteasomes, because no immunoproteasome subunit gene has yet been identified in the *Drosophila* genome.

We have investigated cellular mechanisms involving Stat1, showing that this transcription factor is necessary to induce Psmb9 in the retina as it serves during the cellular response to interferon in the immune system[46]. Stat1 itself is dispensable for retinal development, although it is necessary for accelerating this development in the context of hyperactivated mTORC1, as is also the case for Psmb9 (Figs. 7 and 8). In contrast, deletion of *Stat3* as well as co-deletion of *Stat3* and *Tsc1* in mouse RPCs failed to induce any remarkable differences compared with their respective WT and *Tsc1-cko* mouse retina controls (data not shown), suggesting that *Psmb9* induction is a specific activity of Stat1. However, the mechanism of Stat1 regulation by mTORC1 has been relatively less studied than that of Stat3, which is as a direct target of mTORC1 and supports the proliferation and survival of

neural stem cells[54]. mTOR has been shown to form a complex with Stat1 in response to lipopolysaccharide or IFN-γ and augment Stat1-dependent gene transcription[45]. However, whether mTOR activates Stat1 activity by directly phosphorylating it or whether the mTOR-Stat1 complex induces Psmb9 in the nucleus remains unclear. Thus, further studies are necessary to delineate the mechanistic connection between mTOR and Stat1 in *Psmb9* expression.

Rapamycin is a well-known immunosuppressant that binds to FK506-binding protein, thereby inhibiting calcineurin and subsequent nuclear factor of activated T-cell-dependent production of interleukin-2[55]. Rapamycin also inhibits T-cell activation by interfering with the clonal expansion and antigen-presenting activity of dendritic cells[56]. Our study showed that rapamycin treatment inhibits immunoproteasome β-subunit expression in the adult mouse spleen (Supplementary Fig. 11a), providing an alternative mechanistic interpretation of rapamycin-induced immune suppression. Because the levels of multiple immunoproteasome subunits were increased in the *Tsc1-cko* mouse brain (Supplementary Fig. 11b), future studies should also investigate the potential involvement of the immunoproteasome in various brain disorders, such as autism and epilepsy, caused by mTORC1 hyperactivation.

## Methods

**Mice**. *Tsc1-flox* (*Tsc1^tm1Djk*/J)[24], *Raptor-flox* (B6.Cg-*Rptor^tm1.1Dmsa*/J)[57], *Stat1-ko* (B6.129S(Cg)-*Stat1^tm1Dlv*/J)[58], *Stat3-flox* (*Stat3^tm2Aki*/J)[59], and *R26R* (B6.129S4-*Gt(ROSA)26Sor^tm1Sor*/J)[25] were obtained from Jackson Laboratory, Maine, USA. *Psmb9-ko* mice (B6.129P2-*Psmb9^tm1Stl*>), which were deposited by Dr. Hajime Hisaeda[60], were obtained from RIKEN BioResource Center, Tsukuba, Japan. *Chx10-Cre-GFP* (*Chx10-Cre*) mice were obtained by a generous gift from Dr. Connie Cepko[23]. Experiments using the mice were carried out according to the guidance of KAIST IACUC-2013-54.

**Antibodies and immunostaining**. Mouse embryos were collected from pregnant mice, which were injected with BrdU (5 mg/kg) for the indicated time periods in prior to sample preparation. To inhibit mTORC1 activity in the mouse embryos, rapamycin stock solution (10 mg/ml ethanol) was diluted in vehicle (5% polyethylene glycol and 5% Tween 80 in phosphate-buffered saline (PBS)) for the injection into the intraperitoneal space of the pregnant mice as the indicated schedule in the figures. The embryos were isolated and then fixed in PBS (pH 7.4) with 4% paraformaldehyde (PFA) for 3 h at room temperature. Alternatively, eyes were isolated from the post-natal mice, which were perfused 4% PFA/PBS. The eyes with needle punctures were then incubated in 4% PFA/PBS solution for 16 h at 4 °C. The embryos and eyes were transferred to 20% sucrose/PBS solution until they sank down before subsequent freezing in OCT medium. Immunostaining for cryosections of the embryos and post-natal mouse eyes were performed as reported previously[14]. List of antibodies used for this work were provided in Supplementary Table 1.

**Fluorescence-activated cell sorting (FACS) analysis**. P14 *Tsc1-het* and *Tsc1-cko* adult mouse eyes were dissected and isolated individual retinal cells as described previously[61]. In brief, the cells were incubated in Hank's balanced salt solution (HBSS) containing 4′,6-diamidino-2-phenylindole (DAPI; 1 µg/ml, final) for 10 min, and fluorescent emitted by DAPI-stained dead cells were detected at 405 nm excitation and 452 nm emission using Beckman Courter Flow Cytometer. Relative sizes of DAPI-negative retinal cells were then determined by measuring forward scatter. Relative sizes of BPs were determined by comparing Chx10::GFP-positive cells among DAPI-negative retinal cells analyzed at 495 nm excitation and 519 nm emission.

**Real-time quantitative PCR**. To compare mRNA levels of cell cycle regulators and proteasome subunits, first their cDNA was obtained by reverse transcription with oligo-dT and M-MLV reverse transcriptase (Bioneer). Real-time quantitative PCR (RT-qPCR) was then performed with CFX96 Real-Time System (Bio-Rad) by using iTaq Universal SYBR Green Supermix (Bio-Rad). The RT-qPCR conditions were as follows: 95 °C for 30 s, followed by 40 cycles at 95 °C for 10 s and 60 °C for 30 s, and melt curve stage were 95 °C for 15 s, 60 °C for 1 min, and 95 °C for 15 s. The RT-qPCR was performed with three biological replications, and relative transcript levels of gene of interest were calculated according to the $2^{-\Delta\Delta Ct}$ method against β-actin mRNA. Primer sequences used for the RT-qPCR are provided in Supplemental Table 2.

**[35S]-methionine pulse labeling and chasing in retinal explants**. E13.5 retinal quadrants were incubated in N2-supplemented Dulbecco's modified Eagle's medium (DMEM; Gibco) with 15% fetal bovine serum (FBS) and amino acids but without methionine for 3 h. The [35S]-labeled methionine ([35S]-Met, 250 µCi/ml) was applied to the retinal explants for 3 h. The retinal explants were washed twice with PBS and then added with growth medium containing unlabeled Met (1 mM, final concentration) in the presence and absence of CHX (10 µg/ml). Retinal explants were obtained at time points indicated in the figures, and lysed in a cell lysis buffer (10 mM Tris, 150 mM NaCl, 1% NP40, 1% Triton X-100, and protease inhibitor cocktail [Roche]). Retinal lysates including 0.5 mg proteins were incubated with 1 µg anti-CcnB1 or anti-CcnE1 antibodies at 4 °C for 3 h, and then added with 20 µl (wet volume) protein G-sepharose for 2 h at 4 °C. The immune complexes were centrifuged and washed three times with the cell lysis buffer and twice with the cell lysis buffer with 1 M NaCl. Proteins in the immunoprecipitates were eluted in 20 µl 2× SDS sample buffer for 10% SDS-polyacrylamide gel electrophoresis. The radioactive signal in the gel was then visualized in BAS-7000 (Fuji).

**20S proteasome purification and protease activity assay**. 20S proteasomes were purified from retinal cell lysate in a lysis buffer containing 25 mM HEPES (pH 7.5), 10% glycerol, 5 mM MgCl₂, 1 mM ATP, and 1 mM dithiothreitol by the Proteasome Purification Kit (Enzo Life Sciences BWL-PW9005) following the manufacturer's guide. Immune complex of 20S proteasomes were incubated with the fluorogenic substrate Z-Leu-Leu-Glu-AMC (Enzo, BML-ZW9345), Boc-Leu-Arg-Arg-AMC (Enzo, BML-BW8515), Leu-Leu-Val-Tyr-AMC (Enzo,BML-P802), or Ac-Pro-Ala-Lys-AMC (Boston Biochem Inc.) to measure the fluorescence emitted by cleaved peptide substrates using fluorescence spectroscopy (Tecan Infinite M200) at excitation and emission wavelengths of 380 and 460 nm, respectively, according to the manufacturer's instruction.

**RPC neurosphere culture**. The eyes of E13.5 mouse embryos were enucleated and the retina was dissected out in HBSS with care to avoid contamination from other ocular tissues. Retinas were transferred into a dissociation solution (DMEM containing 0.1% Trypsin and DNAse I (100 µg/ml; Sigma) and incubated at 37 °C in CO₂ incubator for 30 min. The retinas were mechanically dissociated into small aggregates and cultured in DMEM containing 5% FBS for 1 day. The aggregates were then dissociated into single cells using Acumax (Sigma) and grown to form colonies in 0.9%(w/v) methylcellulose matrix (DMEM/F12 containing N2 supplement (GIBCO), B27(GIBCO), 2 mM L-glutamine, 100 U/ml penicillin, and 100 µg/ml streptomycin) supplemented with fibroblast growth factor 2 (10 ng/ml; R&D Systems) and epidermal growth factor (20 ng/ml; Sigma).

**Ubiquitination assay**. The retinas were chopped into small pieces and lysed in a buffer (2% SDS, 150 mM NaCl, 10 mM Tris-Cl, pH 8.0) with protease inhibitor (Calbiochem) and protein phosphatase inhibitor (Sigma) cocktails for 5 min. The cell lysates were boiled for 10 min and then cooled down at room temperature in prior to sonication on ice. Ten volumes of a dilution buffer (10 mM Tris-Cl, pH 8.0, 150 mM NaCl, 2 mM EDTA, and 1% Triton) were added to the cell lysates before the centrifugation at 20 000 × g for 30 min. Lysates including 1 mg proteins were subjected to the immunoprecipitation with antibodies against proteins of interest for subsequent western blot detection of ubiquitinated proteins with mouse anti-ubiquitin antibody.

**Statistical analyses**. Statistical analysis has been performed with SAS software (version 9.4). Group comparison was done using two-sample Student's $t$-test and analysis of variance test, unless it was specified in figure legends. A difference was considered significant at $P < 0.05$.

**Data availability**. The authors declare that all relevant data of this study are included within the article and its Supplementary Information. The data are also available from the authors upon reasonable request.

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

## Acknowledgements

This work was supported by the National Research Foundation of Korea (NRF) grants (2009-00424; 2013M3C7A1056566; and 2017R1A2B3002862) funded by the Korean Ministry of Science, ICT, and Future Planning (MSIP).

## Author contributions

J.-H.C., H.S.J., and J.W.K. wrote the manusucript; J.-H.C., H.S.J., S.L., H.-T.K., K.W.L., K. H.M., T.H., S.S.K., Y.K., and E.J.L. designed and performed the experiments and analyzed the data; C.O.J., and J.W.K. supervised the project.

## Additional information

**Competing interests:** The authors declare no competing interests.

