## [Peer Review File · Nature Communications]

Reviewers' comments:

Reviewer #1 (Remarks to the Author):

In this paper, Choi et al generated a mouse line that specific knock out TSC1 in retina progenitor cells. Starting from this model, they observed that mTORC1 accelerates cell cycle progression. They went on to nicely show that molecularly, mTORC1 supports the synthesis and, surprisingly, also the degradation of Ccn proteins that promote cell cycle progression. The stimulation of Ccn degradation is mainly caused by increasing level of a specific proteasomal protein, Psmb9, which is regulated by the transcription factor Stat1. Please consider the following questions and remarks. Apart from these points, I find that the results are exciting and open up a new direction for further study of mTORC1 function in regulating cell growth and differentiation.

1. The authors should revise the Introduction of this paper. Although I appreciate the effort to thoroughly review the literature, the authors should include a paragraph that clearly spells out the questions to be answered, introduces their methods, and briefly summarizes the results.
2. Figure 2a, the number of Tuj1 positive neurons for "WT + Rapamycin" seems much higher than that of "WT + vehicle", which contradicts the quantification in 2c, as well as the hypothesis that mTORC1 activation leads to accelerated neurogenesis. Is there a switch of panels? If not, the authors should provide an explanation to this result.
3. Figure 3a, the authors should use western blot to directly detect the extent of Raptor knock-out in comparison to the WT line, as a control experiment. The residual pS6 signal raises a concern that the Raptor knock-out may be rather incomplete.
4. The authors should label the cell type in Figure 4g (Tsc1-cKO?).
5. It has previously been established that mTORC1 triggers the translation of a specific type of mRNA (TOP-mRNA, Thoreen ... Sabatini, Nature, 2012). The authors should check whether Psmb5 and Psmb9 belong to this type.
6. The labeling of the cell type in Figure 6a is confusing.
7. The authors should include a model figure to help the readers grasp the essence of their findings.

Reviewer #2 (Remarks to the Author):

In their manuscript titled "mTORC1 accelerates retinal development via the immunoproteasome", Choi et al study the regulatory role of mTORC1 in regulating growth and proliferation of the developing mouse retina. They generated mice with conditional

deletion of Tsc1 in the developing retina using the Chx10-Cre transgene. This led to elevation of mTORC1 activity as measured by pS6. There was an increase in Tuj1 positive cells at E11.5 and this correlated with increased early born cell types. The trend toward early genesis of cell types was recapitulated across development. The authors claim that the proportion of progenitor cells is maintained and that they proliferate more in the Tsc1 knockout using a sphere forming assay. Double labeling experiments were performed at a single timepoint to demonstrate cell cycle progression and increased proliferation in the Tsc1 KO retinae. Inactivation of mTORC1 by in vivo treatment with rapamycin or genetic inactivation of Raptor rescued some of these defects in proliferation and neurogenesis. The levels of cyclin proteins in embryonic mouse retina were higher in the Tsc1 KO and pulse chase experiments suggested that this was not due to increased stability. Indeed, the cyclins were slightly less stable in the Tsc1 KO retinae. The authors went on to show that the proteasome was more active and this was mediated by Psmb9. The genetic inactivation of Psmb9 rescued the phenotype of the Tsc1 KO retinae. Finally, the authors propose that Stat1 is responsible for the regulation of Psmb9 and inactivation of Stat1 also rescues the Tsc1 phenotype.

- 1) The sphere forming assay is not appropriate for showing retinal clonal expansion as retinal progenitor cells do not grow well in such cultures. In vivo clonal analysis is required using retroviruses or other methods.
- 2) Using a single timepoint for the double labeling experiment is not ideal. Multiple timepoints should be used to rule out sustained S-phase. The pH3 co-localization experiment addresses this concern to a certain extent but a few additional timepoints would be important. Also, continuous labeling timecourse is also a very nice way to quantify the proportion of dividing cells and the length of G2+M+G1. See Alexiades and Cepko for an example.
- 3) The analysis of apoptosis in the Raptor KO retinae was only one timepoint and should be repeated for several timepoints.
- 4) The key figures are in the supplemental data and should be moved to the main figure. In particular, the images showing the whole retina by histology and immunostaining in adult and embryonic states need to be in the main figures.
- 5) Those images that are in the supplemental figures do not appear to show any difference to my eye. For example, supplemental Figure 2 all look the same between control and KO. The subtle differences may all be due to angle of cutting the retina (a major issue for those not use to working with the tissue) or artifacts of staining or fixation. These are significant issues for those who do not routinely study retina and should be addressed.
- 6) The difference in cell size should be carefully quantitated using other methods if it is a significant part of the paper.
- 7) The rescue with raptor in supplemental figure 7 does not look convincing. In fact, the H&E in Fig. S7B for the raptor;Tsc1 cKO looks really defective.

Reviewer #3 (Remarks to the Author):

The work by Choi et al, set out to address the role of mTORC1 in coupling neurogenesis to the proliferation of progenitors in vertebrate retinal development. They showed convincingly that, in the absence of Tsc1, the RPCs accelerate both expression and degradation of cell cycle proteins (such as cyclins) that are likely to speed up the cell cycle. They further showed that increase mTORC1 activity induces the expression of the proteasome subunit Psmb9, which partly accounts for the instability of cyclins. This is likely to be mediated by Stat1: co-depletion of both Psmb9 and Stat1 with Tsc1 can rescue most of the precocious differentiation with mildly effect on Cyclins accumulation.

The authors performed extensive analysis and provided an impressive amount of data including some very interesting observations. However, given the established role of mTORC1 in the regulation of growth, the increased proliferation of neuronal progenitors upon mTORC1 upregulation is not particularly surprising, and it remains yet to be determined how the timing or the quantity of the neurons is affected in the cKO of Tsc1 and how it is normally regulated by mTORC1. The potential involvement of Immunoproteasome in neurogenesis is potentially novel. However, the data provided are not convincing enough to support their claims for the role of the immunoproteasome and STAT1 in mediating the increased proliferation of progenitors upon mTORC1 activation. I believe that some major revisions need to be done in order to comply with the Nature Communication standards.

Major concerns:

- The link between the Cyclin accumulation and the immunoproteasome should be explored. First, the authors should perform a similar experiment as the one done on figure sup. 4e, but with co-depletion of Psmb9 or Stat1 to analyse the effect of the immunoproteasome on cell cycle progression. Alternatively, the effect of the immunoproteasome on cyclin stabilisation can be assayed on cell culture.
- Although I am aware of the experimental limitations of the mouse retina, I do not think that the relative proportion to the number of cells is the most convenient way to do it. Specially because most of the figures in the paper do not show the nucleus of all the cells making it difficult for the reader to follow the relative proportions.
- Related to this point, it is not clear in the text and in the description of the phenotype whether conditional knockdown of Tsc1 increases the overall number of cells in the retina rather than speed up the differentiation. The authors should include a quantification of the number of cells in the retina. Moreover, to show that the differentiation is indeed speed up, the authors should analyse time points just before the differentiation is seen in the heterozygous controls.
- Some important experiments in the paper are the quantification data of the protein levels of specific proteins. However, the quality of these experiments is far to comply the standards required for this journal. Specially, the blots of the figures 4b, 5c, 6e (cycB,D), 7b, g, S11 (cycB controls) are frequently overexposed and not suitable for quantification. Blots with less exposed bands should be provided and the quantifications re-done. Also, no error bars are shown in some of the blots and the pulse chase experiments (Fig. 4f-I, 6g-I). How many times were these experiments repeated?
- The authors' argument for the specificity of increased protein stability is mainly based on the autoradiograph date of the total labelled proteins in Sup. Fig. 10. However, as the great majority of proteins are not regulated by proteasomal degradation, this data does not add much information. The authors should address the specificity by either analysing levels of total ubiquitinated proteins, or, blotting some of other known proteasome substrates.

- The authors claim that the cell cycle is faster in the absence of Tsc1, if that is the case how can the authors explain that they observe the differentiation of the late derived neurons (such as bipolar cells and Muller glia), cell types that require longer cell cycle periods. The authors should discuss this.
- The text need to be properly edited as some of the required revisions are pointed out below in my minor points.
- The mitotic index change with time, can it be that mTORC1 slows down the development? An alternative explanation for the increase on mitotic index is that the cells are delayed in mitosis. The authors should clarify this point.
- In Discussion, the authors claim that Stat1 acts downstream of mTORC1 to induce Psmb9. This is clearly an overstatement. No data provided apart from the increase in its protein level (which can be indirect) support that Stat1 acts downstream of mTORC1. This statement should be weakened, or more data should be provided (the authors mentioned that Stat1 and mTOR interact with each other).

Minor points:

- Improve the description of figure S1 and S2 in the main text. It is hard to follow.
- P6 – line 108 , the authors included Crx-positive cone photoreceptors in the timepoint E11.5 and in the figure it is labelled as E14.5. Clarify.
- In figure 1, it is not clear whether the phenotype is due to a general increase in the retina size. The authors mentioned that it is a precocious differentiation but they are using the same time point for both. The ideal experiment would be to identify the appearance of the late-born neurons in early time points. Clarify between timing and size.
- P6, line 118 – Hes1 is not affected. Why do the authors propose that it might affect RPC maintenance by regulating Notch?
- P6, line 125 – Very mild difference in mitotic cells. Cannot explain the big difference in the number of neurons?
- P7, line 142, 146 and 149 – Should be fig. 1g,h instead of d,e)
- Fig. 2 – Increase in PH3 but decrease the BrdU. Why?
- Fig. 2b,d – Quantifications are not representative of the figure. Specially the Wt. Alternatively, does it suggest that the cells are arrested in mitosis for long periods and that might explain why are BrdU negative.
- Sup fig. 7 – The number of Tunel positive is not affected but at the time point analysed, the number of R26+ is not that much affected.
- Fig. 3a – The retinas in some stainings (such as Otx2, Brn3b) are already much depleted from the R26+ cells. Virtually, in these pictures the authors are only showing Wt tissue.
- Split the channels in Fig. 3a. Hard to read.
- Fig. 4d – Graph legend shows control in blue but no values are shown. As it is relative to control, remove this reference.
- Injection of the MG132 intraperitoneal, restore the Ubq levels. Saturating the system? Fig. 11a shows a clear reduction on CycB ubiquitination and that is against the hypothesis of a faster degradation of CycB. MG132 rescue is not convincing.
- P13, line 297 – The authors mention 80% reduction but it should decreased to 80%, rather than decreased by 80%.
- P14, line 314, Psma1 and the other genes mentioned are frequently increased more than 2-folds (fig. 5b) not “less than 2-folds”.
- Fig. 5c – Bad quality blots.

- P14, line 329 – Should be Supplementary figure 12a.
- Fig. 6d,f – green square legend is wrong.
- Fig 6e – low quality blot
- Fig 6g – The bands should be quantified as in figure 4f,g
- P17, line 392 – Should be only fig 7c.
- Fig 7d – third row, the genotype is the same as the fourth row. Mislabelled.
- Fig 7e – the graph has no legend.
- P17, line 393 – the authors mention that no effects are seen in Stat1 KO but it is clear on the graph 7f that absence of Stat1 affect Otx2 and Brnb3 +. Stat1 has effect on its own, doesn't mean that it is downstream of Tsc1.
- P17, line 400, the authors should mention the figure 7g,h
- Different effects on the different subunits should be discussed.

Reviewer #1:

... I find that the results are exciting and open up a new direction for further study of mTORC1 function in regulating cell growth and differentiation.

1. The authors should **revise the Introduction of this paper**. Although I appreciate the effort to thoroughly review the literature, **the authors should include a paragraph that clearly spells out the questions to be answered, introduces their methods, and briefly summarizes the results.**

Re: We rewrote Introduction and added those suggested by the reviewer in the last paragraph.

2. Figure 2a, the number of Tuj1 positive neurons for **“WT + Rapamycin” seems much higher than that of “WT + vehicle”**, which contradicts the quantification in 2c, as well as the hypothesis that mTORC1 activation leads to accelerated neurogenesis. Is there a switch of panels? If not, the authors should provide an explanation to this result.

Re: We replaced the “WT+vehicle” image with another one that can show clear differences from “WT+Rapamycin” image (please see revised Fig.2a).

3. Figure 3a, the authors should **use western blot to directly detect the extent of Raptor knock-out in comparison to the WT line, as a control experiment**. The residual pS6 signal raises a concern that the Raptor knock-out may be rather incomplete.

Re: We provide WB results of Raptor-het and Raptor-cko littermate mouse retinas in Fig.4d.

4. The authors should **label the cell type in Figure 4g (Tsc1-cKO?)**.

Re: We added their genotypes (i.e., WT) in the figure.

5. It has previously been established that **mTORC1 triggers the translation of a specific type of mRNA** (TOP-mRNA, Thoreen ... Sabatini, Nature, 2012). The authors should **check whether Psmb5 and Psmb9 belong to this type**.

Re: We checked that paper carefully, but we could not find Psmb5 and Psmb9 in their TOP-mRNA list.

6. The labeling of the cell type in Figure 6a is confusing.

Re: We provide magnified images in the revised Fig.7a to show the cells of interest more clearly.

7. The authors **should include a model figure to help the readers grasp the essence of their findings.**

Re: We provide a model diagram in Supplementary Fig.14.

Reviewer #2:

1) The **sphere forming assay is not appropriate** for showing retinal clonal expansion as retinal progenitor cells do not grow well in such cultures. **In vivo clonal analysis is required using retroviruses or other methods.**

Re: To provide an evidence for faster expansion of Tsc1-deficient clone than WT clone in same retina, we generated $Tsc1^{ff};TRP1-Cre;R26^{EYFP/+}$ mice by breeding Tsc1-flox and TRP1-Cre mice, in which Cre-affected RPCs are derived from the ciliary margin and differentiate to retinal neurons in isolated clones. We then counted Cre-affected R26EYFP-positive cells in P7 $Tsc1^{+/+};TRP1-Cre;R26^{EYFP/+}$ and $Tsc1^{ff};TRP1-Cre;R26^{EYFP/+}$ littermate mouse retinas. We found not only total numbers of R26EYFP-positive cells but also average number of cells comprising individual R26EYFP-positive clone were elevated in $Tsc1^{ff};TRP1-Cre;R26^{EYFP/+}$ mouse retinas, comparing with those in $Tsc1^{+/+};TRP1-Cre;R26^{EYFP/+}$ littermate retinas. The results therefore suggest that Tsc1-deficient RPCs expand more rapidly than WT RPCs in same retina (Supplementary Fig.3e – h).

2) Using a **single timepoint for the double labeling experiment is not ideal**. Multiple timepoints should be used to rule out sustained S-phase. The pH3 co-localization experiment addresses this concern to a certain extent but a few additional timepoints would be important. Also, continuous labeling timecourse is also a very nice way to **quantify the proportion of dividing cells and the length of G2+M+G1. See Alexiades and Cepko for an example.**

Re: Comparing with the experiments of Alexiades and Cepko who used WT rats, we have a limitation to collect enough numbers of samples. Both control (wt or het) and cko should be obtained from same litter for comparative analyses. Furthermore, we should label BrdU and then chase it for different time points (3h for pH3 co-staining; 12h for Brn3b and Otx2 co-staining; 16h+3h for IdU/CldU co-labeling). To accomplish these experiments at every developmental stage, we should scarify too many mice, and it could not be approved by our IACUC.

We, thus, picked two additional developmental time points, E12.5 and P0, to detect cells entered G2/M from S (i.e., BrdU;pH3 double-positive cells) or completed one round of

cell cycle (i.e., CldU;IdU double-positive cells). Consistent with the results at E14.5, more cells progressed into G2/M from S and completed one round cell cycle in E12.5 and P0 *Tsc1-cko* mouse retinas, respectively, in comparison to those in their *Tsc1-het* littermate mouse retinas. We provide the results for the reviewer's inspection only (please see following results), but do not include into the figures owing to the limited space per figure.

3) The analysis of apoptosis in the Raptor KO retinas was only one timepoint and should be repeated for several timepoints.

Re: We also analyzed Raptor mutant mouse retinas at two additional time points, E12.5 and P0. We found not only total apoptotic cell numbers but also R26R-positive, Raptor-deficient apoptotic cell population were increased in E12.5 and E14.5 Raptor-cko mouse retinas, comparing with those in Raptor-het littermate mouse retinas (Supplementary Fig.6a,e,f). Apoptotic cells in P0 Raptor-cko mouse retinas were also increased, however R26R-positive cells are not majority population of the apoptotic cells (Supplementary Fig.6e,f). Moreover, R26R-positive cells comprise less than 60% in P0 mouse retinas, while they are about 80% in Raptor-het littermate mouse retinas (Supplementary Fig.6d). The results therefore suggest that Raptor deficiency induces RPCs to stop dividing and leads them to apoptotic cell death in embryonic Raptor-cko mouse retina, resulting in the decrease of R26R-positive;Raptor-deficient cell population in the post-natal retina.

4) The **key figures are in the supplemental data and should be moved to the main figure**. In particular, the images showing the whole retina by histology and immunostaining in adult and embryonic states need to be in the main figures.

Re: We moved original Supplementary Fig.2 and 7, which are about mature retinas of Tsc1-cko and Raptor-cko mice, to revised Fig.1 and Fig.4, respectively.

Immunostaining images of whole eye sections for E13.5 and earlier stages are provided in the paper. However, owing to limited space in each figure, we provide only the parts of the retinas at later stages. We provide whole eye section images of later stage samples only for the reviewer's inspection in following (their magnified versions are in Fig.2c), because the staining signals in those whole eye sections are not clearly discriminable at this scale.

5) Those images that are in the supplemental figures do not appear to show any difference to my eye. For example, **supplemental Figure 2 all look the same between control and KO. The subtle differences may all be due to angle of cutting the retina (a major issue for those not use to working with the tissue) or artifacts of staining or fixation.** These are significant issues for those who do not routinely study retina and should be addressed.

Re: We tried to compare the coronal sections including optic head to exclude the positional influences. Please see whole eye section images in the paper (revised Fig.1a; 2a,d; 3a) and those above provided for the reviewer's inspection at above.

As the reviewer indicates, majority of retinal cell types are not significantly different

in their numbers between Tsc1-het and Tsc1-cko samples (originally Supplementary Fig.2, currently Fig.1d,e). However, their sizes are significantly different according to our quantification results (Fig. 1f-i).

6) The difference in **cell size should be carefully quantitated using other methods if it is a significant part of the paper.**

Re: To measure cell size more accurately, we analyzed the sizes of individual cells by FACS (Fig.1h,i). We not only compared the sizes of cells in mixed retinal population of Tsc1-het and Tsc1-cko littermate mouse retinas, but we also compared the sizes of Chx10::GFP-labeled bipolar cells in the retinas. The results clearly show cell size enlargement in Tsc1-cko mouse retinas.

7) The rescue with raptor in supplemental figure 7 does not look convincing. In fact, the **H&E in Fig. S7B for the raptor;Tsc1 cKO looks really defective.**

Re: Co-deletion of Tsc1 could not rescue the phenotypes of Raptor-cko mouse retina completely, although it recovered normal eye size (Fig.4a). Raptor;Tsc1-dko mouse retinas still exhibit rosettes and thinner than Raptor;Tsc1-dhet mouse retinas (Fig.4b,c).

We replaced H&E images in Fig.4b (originally Supplementary Fig.7b). Raptor deletion itself causes retinal hypotrophy and multiple rosettes in the retina. Retinal rosettes are closely related with the absence of Müller glia. Thus, the phenotypes suggest the roles of Raptor in RPC maintenance, which is necessary for the development of late-born retinal cell types including Müller glia in post-natal days.

Reviewer #3:

...The authors performed extensive analysis and provided an impressive amount of data including some very interesting observations. ... I believe that some major revisions need to be done in order to comply with the Nature Communication standards.

Major concerns:

1- The link between the Cyclin accumulation and the immunoproteasome should be explored. First, **the authors should perform a similar experiment as the one done on figure sup. 4e**, but with co-depletion of Psmb9 or Stat1 to analyse the effect of the immunoproteasome on cell cycle progression.

Re: Respecting the reviewer's suggestion, we compared the sizes of neurospheres derived from Tsc1-het, Tsc1-cko, and Tsc1-cko;Psmb9-ko mouse retinal cells, and found the size of neurosphere derived from Tsc1-cko retinal cells was normalized by co-deletion of Psmb9 (Supplementary Fig.12b,c). We also examined mitotic cell cycle progression of RPCs in the neurospheres by detecting BrdU;pH3 double-positive cells. We found the numbers of BrdU;pH3 double-positive cells were also increased remarkably in Tsc1-cko retinal neurospheres, as we observed in vivo, but less significantly in Psmb9-ko;Tsc1-cko retinal neurospheres (Supplementary Fig.12b,d).

Alternatively, **the effect of the immunoproteasome on cyclin stabilisation can be assayed on cell culture.**

Re: The Tsc1-deficient mouse embryonic fibroblasts (MEFs) cannot be maintained (Zhang et al. (2003) J. Clin. Invest. 112:1223–1233). Thus, we analyzed with immortalized p53-/-;Tsc1-/- MEFs, which were transfected with scrambled siRNA or Stat1 siRNA. Knock-down of Stat1 not only decreased levels of Psmb9 but also delayed cell cycle progression of the Tsc1-/- MEFs. To avoid the confusion on our model system (MEFs in culture versus cells in mouse retinas), we do not include the results in the paper, but we provide the results for the reviewer's inspection only (please see following results).

Changes of CcnB1 level during mitotic cell cycle progression of MEFs (Figure for reviewer's inspection only). (a) Immortalized *p53*^{-/-} and *p53*^{-/-};*Tsc1*^{-/-} MEFs were incubated for 16h in the presence of nocodazole (1μM) to arrest their cell cycle at early M-phase, in prior to the incubation in normal growth medium. The MEFs were then collected after the indicated periods, and relative levels of cellular proteins were compared by WB analyses with corresponding antibodies. (b) The *p53*^{-/-};*Tsc1*^{-/-} MEFs were transfected with scrambled-siRNA or Stat1-siRNA at same concentration (20nM), and the cells treated with nocodazole after 12h of transfection.

2- Although I am aware of the experimental limitations of the mouse retina, I do not think that the relative proportion to the number of cells is the most convenient way to do it. Specially because most of the figures in the paper do not show the nucleus of all the cells making it difficult for the reader to follow the relative proportions.

Re: Considering the limited space in each figure, we cannot integrate whole eye section images of late embryonic and post-natal mice at enough resolutions visualizing each individual cells clearly. Thus, we provide magnified versions for the retinas older than E13.5, although we quantified with images covering whole retinal section areas. We provide the images at lower magnification for the reviewer's inspection only (please see enclosed 'Images for reviewer #3's inspection').

3- Related to this point, it is not clear in the text and in the description of the phenotype whether conditional knockdown of Tsc1 increases the overall number of cells in the retina rather than speed up the differentiation. The authors should include a quantification of the number of cells in the retina.

Re: Given the large differences in the numbers of each retinal cell type, the changes of minor retinal cell types, such as RGCs, horizontal cells, and Müller glia, cannot be recognizable in the graphs that set Y-axis scales to cover the numbers of major retinal cell types, such as rod photoreceptors, bipolar cells, and amacrine cells. Therefore, we provide the graphs show relative ratio in the paper. This quantification method has been used in our previous papers including Jo et al. (EMBO J. 31, 817-828), which is cited in the Method.

Moreover, to show that the differentiation is indeed speed up, **the authors should analyse time points just before the differentiation is seen in the heterozygous controls.**

Re: By respecting the reviewer's suggestion, we examined the expression of those markers in the earlier stages (at E10.5 for Calbindin and Isl1; at E12.5 for Crx; at E16.5 for Rhodopsin; at P0 for Lhx3). We could found those retinal cell markers in those Tsc1-cko mouse retinas at earlier time points, when the markers are undetectable (or seen very sparsely) in Tsc1-het littermate mouse retinas. We replaced original results with these at earlier stages (Fig.2c).

4- Some important experiments in the paper are the quantification data of the protein levels of specific proteins. However, **the quality of these experiments is far to comply the standards required for this journal. Specially, the blots of the figures 4b, 5c, 6e (cycB,D), 7b, g, S11 (cycB controls) are frequently overexposed and not suitable for quantification.**

Re: We replaced those blots mentioned by the reviewer with those in better qualities.

Blots with less exposed bands should be provided and the quantifications re-done. Also, **no error bars are shown in some of the blots and the pulse chase experiments (Fig. 4f-l, 6g-l). How many times were these experiments repeated?**

Re: We added the error bars in the revised graphs. The numbers of experiments are provided in corresponding figure legends.

5- The authors' argument for the specificity of increased protein stability is mainly based on the autoradiograph data of the total labelled proteins in Sup. Fig. 10. However, as the great majority of proteins are not regulated by proteasomal degradation, this data does not add much information. **The authors should address the specificity by either analysing levels of total ubiquitinated proteins, or, blotting some of other known proteasome substrates.**

Re: We examined relative levels of total ubiquitinated proteins and other ubiquitinated proteins, such as β -actin and p53, in Tsc1-het and Tsc1-cko littermate mouse retinas. We could not find significant differences between those two retinal samples (Supplementary Fig. 10f).

6- The authors claim that the cell cycle is faster in the absence of Tsc1, if that is the case **how can the authors explain that they observe the differentiation of the late derived neurons (such as bipolar cells and Muller glia), cell types that require longer cell cycle periods.** The authors should discuss this.

Re: To understand the relationship between cell cycle and retinal histogenesis, Edward Levine and colleagues investigated the roles of CcnD1 and p27 in mouse retinal development. It has been known that CcnD1 not only supports cells to entering cell cycle but also prevents them from exiting cell cycle. Consistently, CcnD1-ko embryonic RPCs exit cell cycle prematurely and are gradually depleted in the retina (Das et al. Neural Development 4, 15). This results in the increase of first-born retinal cell-type RGC, but the decrease of next coming retinal cell-types, such as horizontal cells and amacrine cells. Despite the faster depletion of RPC population in the embryonic stages, RPC population remain in the post-natal CcnD1-ko mouse retina and they even produce late-born retinal types, such as bipolar cells and Muller glia, precociously (Das et al. Developmental Dynamics 241, 941–952). The opposing roles of CcnD1 in RPC cell cycle in embryonic and post-natal stages are explained by the authors as “Ccnd1 influences the timing of cell-cycle exit in subsets of RPCs with limited proliferative potential”. It means that CcnD1 do not always inhibit cell cycle exit of RPC, but it sometimes promote it. Thus, the precociously produced Muller glia might be due to fast cell cycle entry of CcnD1-ko RPC sup-population but not due to the extended cell cycle length. The p27 is necessary for

RPCs to exiting cell cycle by antagonizing CcnD1/Cdk4 activation. Thus, p27-deficient RPCs are capable of retinal histogenesis exceeding a regular schedule (Levine et al., Developmental Biology 219, 299–314). However, retinal composition is unaffected in p27-deficient mouse retina. Collectively, these results suggest that the length of cell cycle is not directly linked to fate determination for specific retinal neurons and Muller glia.

It was suggested that fate determination of RPC during development is stochastic (Gomes et al., Development 138, 227-235; He et al., Neuron 75, 786-798). In the stochastic model, RPC expands to form a clone, in which majority of retinal cells are produced although the order does not strictly follow the well-known RGC->Cone/HZ/AC->rod->BP->MG flow. Therefore, composition of RPC-derived clone might not be different significantly each other, regardless of different speed of neurogenesis in each clone. This means that cell composition of Tsc1-cko mouse retina should not be different significantly from Tsc1-het mouse retina, even though Tsc1-cko RPCs divide faster and completed retinal histogenesis earlier than Tsc1-het littermates.

Minor points:

- The mitotic index change with time, can it be that mTORC1 slows down the development?

An alternative explanation for the increase on mitotic index is that the cells are delayed in mitosis. The authors should clarify this point.

Re: We measured CldU/IdU double-labeled cells to determine whether the elevated pH3-positive cells in Tsc1-deficient RPCs was resulted from delay in mitosis or a result of faster mitotic cell cycle progression. The results suggest that Tsc1-deficient CldU-labeled RPCs entered next S-phase to incorporate IdU earlier than Tsc1-het RPCs (Fig.2f,g). Therefore, RPCs might have completed mitosis to enter next cell cycle faster than regular speed.

Alternatively, as we show in the MEF analysis, onset time for CcnB1 and pH3 accumulation in p53^{-/-};Tsc1^{-/-} MEFs were earlier than p53^{-/-} MEFs. Finally, we provide a theoretical model in Supplementary Fig.13a, which explains why CcnB1 is detectable at higher level in the retina comprised of RPCs with fast cell cycle than the normal retina. Collectively, we think increased mitotic index in Tsc1-deficient cells is not resulted from

delayed mitosis but rather from accelerated cell cycle progression.

- In Discussion, the authors claim that Stat1 acts downstream of mTORC1 to induce Psmb9. This is clearly an overstatement. No data provided apart from the increase in its protein level (which can be indirect) support that Stat1 acts downstream of mTORC1. **This statement should be weakened**, or more data should be provided (the authors mentioned that Stat1 and mTOR interact with each other).

Re: We rewrote the sentence as "...showing that this transcription factor is necessary to induce Psmb9 in the retina as it serves during the cellular response to interferon in the immune system" (page 21, line 512-513).

- Improve the description of figure S1 and S2 in the main text. It is hard to follow.

Re: We added the explanations about Fig.1 (previously Supplementary Fig.2) and Supplementary Fig.1 in the main text (page 5-6, line 93-105).

- P6 – line 108, the authors included Crx-positive cone photoreceptors in the timepoint E11.5 and in the figure it is labelled as E14.5. Clarify.

Re: It was E14.5. We replaced the data with those of E12.5 in the revised Fig.2c.

- In figure 1, it is not clear whether the phenotype is due to a general increase in the retina size. The authors mentioned that it is a precocious differentiation but they are using the same time point for both. **The ideal experiment would be to identify the appearance of the late-born neurons in early time points.** Clarify between timing and size.

Re: We examined the presence of retinal neurons in the earlier time points than those we tested in the original version (Fig.2c,d). Among the late-born neurons, we could see Lhx3-positive bipolar cells at higher numbers in P0 Tsc1-cko mouse retinas than those in Tsc1-het littermate mouse retinas, but we could not see them in the embryonic mouse retina regardless of genotypes (data not shown).

- P6, line 118 – Hes1 is not affected. Why do the authors propose that it might affect RPC maintenance by regulating Notch?

Re: In our previous paper (Jo et al. (2012) EMBO J 31, 817-828), we showed the inactivation of Notch signaling by Akt. Thus, we wondered whether the activation of mTORC1 via the phosphorylational inactivation of Tsc2 by Akt could also affect Notch signaling to promote retinal neurogenesis, but we could not find the difference between Tsc1-het and Tsc1-cko mouse retinas. We rewrote it as “However, RPC population, which can be marked by Vsx2 and Hes1, is well preserved in Tsc1-cko mouse retinas (Supplementary Fig.2e,f)” in the revised text (page 7, line 132-136).

- P6, line 125 – Very mild difference in mitotic cells. Cannot explain the big difference in the number of neurons?

Re: Given the increased mitotic index, Tsc1-cko embryonic RPCs can produce more neurons than Tsc1-het littermates during same time period. If this is repeated during development, significantly larger numbers of retinal cells will be produced in Tsc1-cko mouse retinas than those in Tsc1-het littermate retinas.

- P7, line 142, 146 and 149 – Should be fig. 1g,h instead of d,e)

Re: We corrected those in the revised text (page 8, line 159, 163, 166).

- Fig. 2 – Increase in PH3 but decrease the BrdU. Why?

Re: In rapamycin-treated mouse retinas, increase of pH3 is a reflection of mitotic cell cycle arrest, and decrease of BrdU represents reduced entry to S-phase from G1. These are different from the cases of Tsc1-cko mouse retinas, in which elevated BrdU;pH3-positive cells represent RPCs progress rapidly to G2/M-phase after completing S-phase.

- Fig. 2b,d – Quantifications are not representative of the figure. Specially the Wt.

Re: We replaced with other images, which are more representing the quantification (revised Fig.3b,d).

Alternatively, does it suggest that the cells are arrested in mitosis for long periods and that might explain why are BrdU negative.

Re: Yes, we think many rapamycin-treated RPCs cannot enter S-phase by being arrested at G0/G1 phase and/or mitotic phase (please see the diagram in Fig.4h).

- Sup fig. 7 – The number of Tunel positive is not affected but at the time point analysed, the number of R26+ is not that much affected.

Re: We found total TUNEL-positive cells per section was also larger in E14.5 Raptor-cko mouse retinas, after we repeated analyses during revision. Moreover, they are more concentrated to R26R-positive Cre-affected cell population than R26R-negative wild-type cell population in Raptor-cko embryonic mouse retinas. It might result in the depletion of R26R-positive Raptor-deficient cells in the post-natal Raptor-cko mouse retinas (please the results in Supplementary Fig.6).

- Fig. 3a – The retinas in some stainings (such as Otx2, Brn3b) are already much depleted from the R26+ cells. Virtually, in these pictures the authors are only showing Wt tissue.

Re: The results are related with the depletion of Raptor-deficient mouse RPCs. The inactivation of mTORC1 in Raptor-deficient RPC might result in cell cycle arrest and the death of the cells (Supplementary Fig.6). Consequently, they were not only able to renew themselves, but also fail to produce retinal neurons, resulting in relative dominance of wild-type cell population, which are not depleted and expand at regular speed, in the Raptor-cko mouse retina.

- Split the channels in Fig. 3a. Hard to read.

Re: We separated color channels of the markers from R26R, and provided the images in the revised Fig.4e.

- Fig. 4d – Graph legend shows control in blue but no values are shown. As it is relative to control, remove this reference.

Re: We removed the blue box, and relabeled other boxes.

- Injection of the MG132 intraperitoneal, restore the Ubq levels. Saturating the system? Fig. 11a shows a clear reduction on CycB ubiquitination and that is against the hypothesis of a faster degradation of CycB. MG132 rescue is not convincing.

Re: We tested with MG132 at various concentrations and identified the saturated

concentration (data not shown).

As the reviewer indicated, given the elevated CcnB1 level in Tsc1-cko mouse retina, the levels of ubiquitinated-CcnB1 is also likely higher in the retina than Tsc1-het mouse retina. However, proteasome activity in Tsc1-cko mouse retina is also elevated, and ubiquitinated-CcnB1 will be degraded more rapidly in Tsc1-cko mouse retinas than in Tsc1-het mouse retinas. Therefore, less ubiquitinated-CcnB1 protein could be left in Tsc1-cko mouse retinas than that in Tsc1-het mouse retinas, although total CcnB1 level is higher in the retina.

However, the levels of ubiquitinated-CcnB1 are not significantly different between MG132-treated Tsc1-het and Tsc1-cko littermate mouse retinas. Furthermore, the difference of ubiquitinated-CcnB1 between MG132-treated and -untreated Tsc1-cko mouse retinas is larger than the difference in Tsc1-het mouse retinas (revised Supplementary Fig.10e), reaffirming the efficiency of proteasomal degradation of ubiquitinated-CcnB1 is higher in Tsc1-cko mouse retina.

- P13, line 297 – The authors mention 80% reduction but it should decreased to 80%, rather than decreased by 80%.

Re: We corrected it (page 14, line 321).

- P14, line 314, Psma1 and the other genes mentioned are frequently increased more than 2-folds (fig. 5b) not “less than 2-folds”.

Re: Psma1 and Psmc1 mRNA levels in Tsc1-cko samples are increased about 2-folds in comparison to Tsc1-het sample, but the catalytic subunits of constitutive proteasome (Psmc1, b2, b5) was increased less than 2-folds. We corrected the expression in the revised text (page 15, line 338-339).

- Fig. 5c – Bad quality blots.

Re: We repeated the experiments and replaced with those in better band qualities (revised Fig.6c).

- P14, line 329 – Should be Supplementary figure 12a.

Re: It is Supplementary Fig.12b (related to Emx2-Cre). We corrected this in the revised text (page 15, line 358).

- Fig. 6d,f – green square legend is wrong.

Re: We labeled it correctly, and also fixed those in Fig.7e (WB).

- Fig 6e – low quality blot

Re: We replaced those in better qualities.

- Fig 6g – The bands should be quantified as in figure 4f,g

Re: We quantified those in the revised Fig.7h,i.

- P17, line 392 – Should be only fig 7c.

Re: We corrected it in the revised Fig.8c.

- Fig 7d – third row, the genotype is the same as the fourth row. Mislabelled.

Re: We corrected it in the revised Fig.8d.

- Fig 7e – the graph has no legend.

Re: We add labels in the revised Fig.8e.

- P17, line 393 – the authors mention that no effects are seen in Stat1 KO but it is clear on the graph 7f that absence of Stat1 affect Otx2 and Brnb3 +. Stat1 has effect on its own, doesn't mean that it is downstream of Tsc1.

Re: We repeated experiments and counted more samples to modify the graph, which shows no significant increase in the numbers of Otx2- and Brn3-positive retinal neurons in Tsc1-cko;Stat1-ko mouse retinas (revised Fig.8f).

- P17, line 400, the authors should mention the figure 7g,h

Re: We matched the part in the first sentence with Fig 8g,h (page 18, line 436-437).

- Different effects on the different subunits should be discussed.

Re: We mentioned this in the part discussing about the effects of immunoproteasomes in the brain (page 22, line 435-439).

Reviewers' comments:

Reviewer #2 (Remarks to the Author):

The authors have addressed my concerns and suggestions.

Reviewer #3 (Remarks to the Author):

I acknowledge the effort the authors put into this revision. I acknowledge that the revised manuscript has addressed all my concerns and that the overall quality of the manuscript has significantly improved. I am now convinced that the provided data support the main claims of the authors and that this study contains the novelty and the impact that the journal's readership would expect.

However, there are still the following minor, yet critical points that need attention. For this reviewer to fully support publication of this manuscript, these points should be fully addressed.

1. In the introduction, the background of the TOR pathway and its known developmental function, which is the main topic of this manuscript, is insufficient for non-expert readers.
2. In Page5 Line100, the term "P14" should be defined for general readers at its first appearance (the same as "E14..5").
3. In P7 L137, as above, ROSA26lacZ should be explained.
4. In Figure S3 and S13, turn the figure 90°anti-clock-wise.
5. In P9 L188, no explanation for the H3P increase upon rapamycin treatment is given. This increase in H3P and the decrease in the ratio of BrdU+ cells in the H3P positive population indicates a delay in the progression into mitosis. This should be clearly stated (as stated in the rebuttal letter).
6. In P10 L210, please remove "potentially".
7. An explanation for the phenotypic difference between Rapamycin-treated retina and Raptor-cko retina, in particular, in regard to the occurrence of massive cell death in Raptor-cko retina, should be given.
8. Figure 4h is uninterpretable to this reviewer. I would recommend to either revise or remove this figure.
9. In P11 L243-244, CyclinA and Cyclin E both bind to CDK2, not CDK4.
10. In Fig. 5a, the authors show that CDK inhibitor p27 was also increased in Tsc1-cko, which is not consistent with the accelerated cell cycle in this condition. The authors should explain this.
11. In Page 12 Line258-261, CHX just shut down translation completely. The interpretation, "the effects of rapamycin...of translation", does not make any sense to me. The authors should remove this part or explain the logic.
12. In Page 13 L287, related to Fig S9, proper quantification should be done for S35-labeled total proteins (the degradation rates of Cyclin B and Cyclin E were also not obvious without quantification).
13. In Fig.S10f, proper quantification of the signal using unsaturated blots, as in S10b, is required.
14. In P14 L310-311, MG132 treatment clearly further accumulated ubiquitinated Cyclin B1

in the rapamycin-treated retina in Fig.S10c (the a-Ubi signal in Lane 2 looks more than 2 times as high signal intensity as that in Lane 4), but the graph S10d does not seem to reflect it. This makes me doubtful about the quantification method the authors used. The authors should explain the discrepancy.

15. In P15 L354, Psmb6/B1i and Pam7/b2i should be Psmb6/B1 and Pam7/b2.

16. In P15 L356-358, does the increase in the other immunoproteasome subunits in the cerebrum have any implications in their regulation in the retina? If no clear reason is given, this should be removed.

17. P16 L375, "FigS12a third ... rows" should be FigS12a second.

18. In P17, since the model suggests that Tsc1-cko mice degrade cyclin accumulation more efficiently by the upregulation of Psmb9, it is expected that additional knockout of Psmb9 in this background accumulates Cyclins even further. However, the Western blots show the opposite. This indicates that changes in cyclin levels are not good predictors of the speed of cell cycle progression by TOR. Considering that the cyclin levels are affected by many factors (the states of the cell cycle, transcription/translation rates, protein stability), it is understandable. The authors should take this point into account and should give more equal discussions on their data, not only mentioning the part of data that fits their hypothesis. Furthermore, whereas the "timing" of cyclin destruction is critical for cell cycle progression, there is no evidence that the "rates" of cyclin degradation affect the speed of the cell cycle.

19. Fig. S13 shows a theoretical model, which the authors use to explain the observed effects on the cyclin levels. Thus, if it is valid, it is supposed to be an important piece of data and the result should be properly described in the main text. However, in the current state of this data, I am not convinced that this model is valid according to the description in the figure legend. What is the "guess" and what evidence is this assumption of the 30 hours of the length of the RPC cell cycle based upon? This model also does not include the observed changes in transcription and translation rates of cyclins upon tsc1 knockout or rapamycin treatment. To me, it seems as though the authors had chosen the parameters just to produce the outputs that would fit their data. The authors should remove this figure entirely, or provide a thorough explanation. In addition, the quality of the images in a,d, g is too poor and the labels are unreadable, so in the latter case, the authors should provide the images in higher quality.

Reviewer's comments: black (key points are underlined)

Authors' responses to the reviewer's comments: *blue italic*

Reviewer #3

I acknowledge the effort the authors put into this revision. I acknowledge that the revised manuscript has addressed all my concerns and that the overall quality of the manuscript has significantly improved. *I am now convinced that the provided data support the main claims of the authors and that this study contains the novelty and the impact that the journal's readership would expect.*

However, there are still the following minor, yet critical points that need attention. For this reviewer to fully support publication of this manuscript, these points should be fully addressed.

1. In the introduction, the background of the TOR pathway and its known developmental function, which is the main topic of this manuscript, is insufficient for non-expert readers.

-> *We added an introduction of mTOR pathway and its roles in development (page 2, highlighted).*

2. In Page5 Line100, the term "P14" should be defined for general readers at its first appearance (the same as "E14..5").

-> *We provide definitions for P14 (post-natal day 14) and E11.5 (embryonic day 11.5) in the revised text (page 6, highlighted).*

3. In P7 L137, as above, ROSA26lacZ should be explained.

-> *We provide the explanation for ROSA26lacZ (R26R) in the revised text (page 7, highlighted).*

4. In Figure S3 and S13, turn the figure 90°anti-clock-wise.

-> *We rotated the figures.*

5. In P9 L188, no explanation for the H3P increase upon rapamycin treatment is given. This increase in H3P and the decrease in the ratio of BrdU+ cells in the H3P positive population indicates a delay in the progression into mitosis. This should be clearly stated (as stated in the rebuttal letter).

-> *We added the interpretation of those results in the revised text (page 10, highlighted).*

6. In P10 L210, please remove "potentially".

-> *We removed it.*

7. An explanation for the phenotypic difference between Rapamycin-treated retina and Raptor-cko retina, in particular, in regard to the occurrence of massive cell death in Raptor-cko retina, should be given.

-> *The effects of Raptor deletion were accumulated for more than 4 days in the Raptor-cko mouse retina, considering the onset time (i.e., ~ E10) of Chx10-Cre (Rowan and Cepko, 2004). However, rapamycin affected the mouse embryos as long as 16 hours. Therefore, mTORC1 inhibition by rapamycin might not be long enough to induce massive apoptosis in the retina. Unfortunately, extended treatment by repeated injection of rapamycin killed the embryos, thus we could not examine the effects for longer period.*

8. Figure 4h is uninterpretable to this reviewer. I would recommend to either revise or remove this figure.

-> *We removed the diagram (please see revised Fig4).*

9. In P11 L243-244, CyclinA and Cyclin E both bind to CDK2, not CDK4.

-> *We corrected it in the revised text (page 12, highlighted).*

10. In Fig. 5a, the authors show that CDK inhibitor p27 was also increased in Tsc1-cko, which is not consistent with the accelerated cell cycle in this condition. The authors should explain this.

-> The p27 is expressed in post-mitotic cells in mouse embryonic retina (Levine et al., 2000; Dyer and Cepko, 2000). Given the significant increase of post-mitotic cells in Tsc1-cko mouse retinas comparing with Tsc1-het littermates (Fig2), p27 elevation could be accounted by the increase of post-mitotic cells in the retina.

11. In Page 12 Line258-261, CHX just shut down translation completely. The interpretation, “the effects of rapamycin....of translation”, does not make any sense to me. The authors should remove this part or explain the logic.

-> It has been known that rapamycin suppresses translation by inhibiting mTORC1 (Ma and Blenis, 2009). We, thus, wondered if the elevation of cyclin proteins in rapamycin-treated mouse retina was caused by the suppression of cyclin mRNA translation and tried to get the answer by blocking translation by CHX. We added the background in the text (page 12, highlighted).

12. In Page 13 L287, related to Fig S9, proper quantification should be done for S35-labeled total proteins (the degradation rates of Cyclin B and Cyclin E were also not obvious without quantification).

-> We provide relative intensities of [35S]-Met-labeled proteins in each lane below the autoradiography images (please see revised FigS9).

13. In Fig.S10f, proper quantification of the signal using unsaturated blots, as in S10b, is required.

-> We provide the quantification in a graph (FigS10g).

14. In P14 L310-311, MG132 treatment clearly further accumulated ubiquitinated Cyclin B1 in the rapamycin-treated retina in Fig.S10c (the a-Ubi signal in Lane 2 looks more than 2 times as high signal intensity as that in Lane 4), but the graph S10d does not seem to reflect it. This makes me doubtful about the quantification method the authors used. The authors should explain the discrepancy.

-> *We re-quantified the data with Image-J and Photoshop programs, and obtained 1.79 and 1.82-fold increases, respectively, in Tsc1-cko mouse retinas. Those values are not quite different from that in the graph.*

15. In P15 L354, Psmb6/B1i and Pam7/b2i should be Psmb6/B1 and Pam7/b2.

-> *We corrected those in the revised text (page 16, highlighted).*

16. In P15 L356-358, does the increase in the other immunoproteasome subunits in the cerebrum have any implications in their regulation in the retina? If no clear reason is given, this should be removed.

-> *Regulation of immunoproteasome subunit expression might be different between tissues. Expression of immunoproteasome subunits in the cerebrum and spleen might be regulated by other transcription factors, which are also sensitive to mTORC1 activity, besides Stat1. Therefore, upon deletion of Tsc1, all three immunoproteasome-specific subunits can be elevated in the cerebrum and spleen, whereas only Psmb9 is induced in the retina.*

17. P16 L375, “FigS12a third rows” should be FigS12a second.

-> *It is presented properly. The second row of FigS12a shows tubulin beta-III stained by Tuj1, whereas the sentence is related to BrdU- and pH-positive cells.*

18. In P17, since the model suggests that Tsc1-cko mice degrade cyclin accumulation more efficiently by the upregulation of Psmb9, it is expected that additional knockout of Psmb9 in this background accumulates Cyclins even further. However, the Western blots show the opposite. This indicates that changes in cyclin levels are not good predictors of the speed of cell cycle progression by TOR. Considering that the cyclin levels are affected by many factors (the states of the cell cycle, transcription/translation rates, protein stability), it is understandable. The authors should take this point into account and should give more equal discussions on their data, not only mentioning the part of data that fits their hypothesis. Furthermore, whereas the “timing” of cyclin destruction is critical for cell cycle

progression, there is no evidence that the “rates” of cyclin degradation affect the speed of the cell cycle.

-> *Our results show not only the synthesis but also the degradation of cyclin proteins is enhanced in Tsc1-cko mouse retina (Fig5). Given the oscillation of cyclin levels during cell cycle, mTORC1 should coordinately accelerate synthesis and degradation of cyclin proteins to promote RPC proliferation. We found the mTORC1-induced enhancement of cyclin synthesis occurs at both transcription (FigS8) and translation (Fig5f) levels, although we should wait for future studies that will give an answer which process is more sensitive to mTORC1 activation. Importantly, we show the degradation of [35S]-Met-labeled cyclin proteins is accelerated in Tsc1-cko mouse retina without significant increase in its ubiquitination rate (Fig5e-i; FigS10). This suggests that enhanced proteasomal degradation of ubiquitinated cyclins contributes more significantly to the accelerated turnover of cyclins in Tsc1-cko mouse retina than the ubiquitination itself. We found Psmb9 was induced most significantly among proteasome subunits in Tsc1-cko mouse retina, implicating that Psmb9 might be responsible for the acceleration of cyclin turnover in the retina. Decay of [35S]-Met-labeled cyclin proteins was delayed in Psmb9-ko;Tsc1-cko mouse retina comparing with that in Tsc1-cko mouse retina (Fig7g-i). This suggest the important roles of Psmb9-containing immunoproteasomes in accelerating cyclin decay. These were simplified in the theoretical models in previous FigS13, which is removed in this version. However, we believe all these points were summarized in corresponding parts in the revised text, thus our reviewers may be able to figure those points out without difficulty.*

We agree to reviewer’s opinion that the timing of cyclin destruction is as much important as the rate. It should be noted that the timing of cyclin destruction is determined by factors accumulated in past cell cycle phases, whereas the rate is sensitive to the contents of destruction machineries, such as ubiquitinating enzymes and the proteasomes. Majority of regulators, including Cdk1-CcnB and APC/C-Cdc20, identified by far are related with destruction timing. However, the factors regulating destruction rate are not known well. In this study, we propose the immunoproteasome (more specifically Psmb9 in the retina) as a regulator of cyclin destruction rate.

19. Fig. S13 shows a theoretical model, which the authors use to explain the observed effects on the cyclin levels. Thus, if it is valid, it is supposed to be an important piece of data and the result should be properly described in the main text. However, in the current state of this data, I am not convinced that this model is valid according to the description in the figure legend. What is the “guess” and what evidence is this assumption of the 30 hours of the length of the RPC cell cycle based upon? This model also does not include the observed changes in transcription and translation rates of cyclins upon *tsc1* knockout or rapamycin treatment. To me, it seems as though the authors had chosen the parameters just to produce the outputs that would fit their data. The authors should remove this figure entirely, or provide a thorough explanation. In addition, the quality of the images in a,d, g is too poor and the labels are unreadable, so in the latter case, the authors should provide the images in higher quality.

-> As the reviewer indicated, the model is simplified to fit into our results. We thought it could help our readers to understand the results in the paper better. We provide a schematic diagram in revised Fig.S12, thus it can be removed now.

REVIEWERS' COMMENTS:

Reviewer #3 (Remarks to the Author):

In the revised manuscript, the authors have properly addressed all the concerns raised. Thus, I now recommend this manuscript for publication.

In regard to the authors response to point 11, I don't quite understand the logic behind this explanation: how can the inhibition of cyclin mRNA translation elevate their protein levels? However, the actual text, newly added in p12, in the revised manuscript, "rapamycin treatment.... by interfering with the synthesis of proteins that inhibit cyclin synthesis or promote cyclin decay", does indeed make more sense. Although I don't think that the data of the CHX treatment does not rule out this particular hypothesis, it can support the notion that cyclin accumulation is, at least, not due to the inhibition of general translation.

Our response to the reviewer's comment

In regard to the authors response to point 11, I don't quite understand the logic behind this explanation: how can the inhibition of cyclin mRNA translation elevate their protein levels? However, the actual text, newly added in p12, in the revised manuscript, "rapamycin treatment... by interfering with the synthesis of proteins that inhibit cyclin synthesis or promote cyclin decay", does indeed make more sense. Although I don't think that the data of the CHX treatment does not rule out this particular hypothesis, it can support the notion that cyclin accumulation is, at least, not due to the inhibition of general translation.

Response to the comment: *We do not mean that inhibition of cyclin mRNA translation elevates their protein levels. Rapamycin has been known as an inhibitor of mRNA translation, which is sensitive to mTORC1 activation. It was therefore surprising for us to observe the elevation of cyclin protein levels without corresponding increase of their mRNA levels. Thus, as we wrote in the text, we thought rapamycin might increase cyclin protein level indirectly by inhibiting the synthesis of the proteins that inhibit cyclin synthesis or promote cyclin decay. To test this possibility, we treated a general translation inhibitor cycloheximide (CHX) to the embryos. However, as opposite to rapamycin treatment, CHX treatment decreased the levels of cyclin proteins. Therefore, we concluded that rapamycin did not increase cyclin levels via translation inhibition but might increase in other means. It intrigued us to test whether rapamycin inhibit the decay of cyclin proteins. We rewrote the text more detail as "Given the negative effects of rapamycin on translation³⁷, the results were quite surprising. We thus hypothesized that rapamycin treatment might accumulate cyclin proteins by interfering with the synthesis of the proteins that inhibit cyclin synthesis or promote cyclin decay. However, in opposite to the effects of rapamycin, the levels of cyclin proteins were commonly decreased in embryonic retinas isolated from mice injected with a general translation inhibitor cycloheximide (CHX; Fig.5c,d). These results therefore imply that cyclin accumulation in rapamycin-treated mouse retina was not resulted from the inhibitory effects of rapamycin on translation."*